# WHERE IN THE WORLD? A VISION-LANGUAGE BENCHMARK FOR PROBING MODEL GEOLOCATION SKILLS ACROSS SCALES

## ABSTRACT

Vision-language models (VLMs) have advanced rapidly, yet their capacity for image-grounded geolocation in open-world conditions, a task that is challenging and of demand in real life, has not been comprehensively evaluated. We present `WhereBench`, a comprehensive benchmark for VLM image geolocation that evaluates visual recognition, step-by-step reasoning, and evidence use. `WhereBench` comprises 810 globally distributed images across two complementary geolocation scales: `WhereCountry` (*i.e.*, 500 multiple-choice question-answering, with country-level answer and panoramas) and `WhereStreet` (*i.e.*, 310 fine-grained street-level identification tasks requiring multi-step reasoning with optional web search). For evaluation, we adopt the final-prediction metrics: location accuracies within $k$ km ($Acc@k$) for coordinates and hierarchical path scores for textual localization. Beyond this, we propose to explicitly score intermediate reasoning chains using human-verified key visual clues and a Shapley-reweighted thinking score that attributes credit by each clue's marginal contribution. We benchmark 12 state-of-the-art VLMs with web searching tools on our `WhereBench` and report different types of final answer accuracies as well as the calibrated model thinking scores. We reveal that web search and reasoning do not guarantee improved performance when visual clues are limited, achieving only an overall 56.3% with the best SOTA model. These findings highlight not only the promise but also the persistent challenges of models to mitigate bias and achieve robust, fine-grained localization.

## 1 INTRODUCTION

Vision–language models (VLMs) have advanced multimodal perception and decision making, enabling AI systems to reason over images and, when necessary, invoke external tools such as image editing or web search to tackle tasks with deeper understanding and stronger capabilities (Qi et al., 2024; Zheng et al., 2025; OpenAI, 2025b;a; Team et al., 2025). Image geolocation serves as a natural testbed for vision-grounded reasoning and tool using: given an image, the goal is to infer its location or coordinates. This capability matters in practice, such as search and rescue (Kim et al., 2021), urban planning (Glistrup et al., 2022), or environmental monitoring (Lotfian and Ingensand, 2021). Meanwhile, this paradigm is different from conventional VLM benchmarks that put their primary focus on model capacities for difficult question-answering. However, there remains a lack of a fair and comprehensive benchmark that evaluates not only final localization accuracy but also the faithfulness of the underlying reasoning process.

Solving image geolocation tasks requires careful analysis of visual cues (*e.g.*, signs, architecture, vegetation), retrieval of corroborating evidence, and synthesis into a final prediction. Recent VLM evaluations predominantly target general multimodal capabilities (Cheng et al., 2025; Lin et al., 2025; Lee et al., 2024; Li et al., 2024a), focusing on perception, reasoning, and safety, while neglecting other dimensions such as localization from limited information. The localization task is inherently difficult even for human because it requires either extensive knowledge covering the image content or strong tool-use abilities (Wazzan et al., 2024) to search for external knowledge from visual cues. While there are previous works evaluating localization settings (Vo et al., 2017; Clark et al., 2023; Huang et al., 2025), they are conducted under isolated settings where external tools and

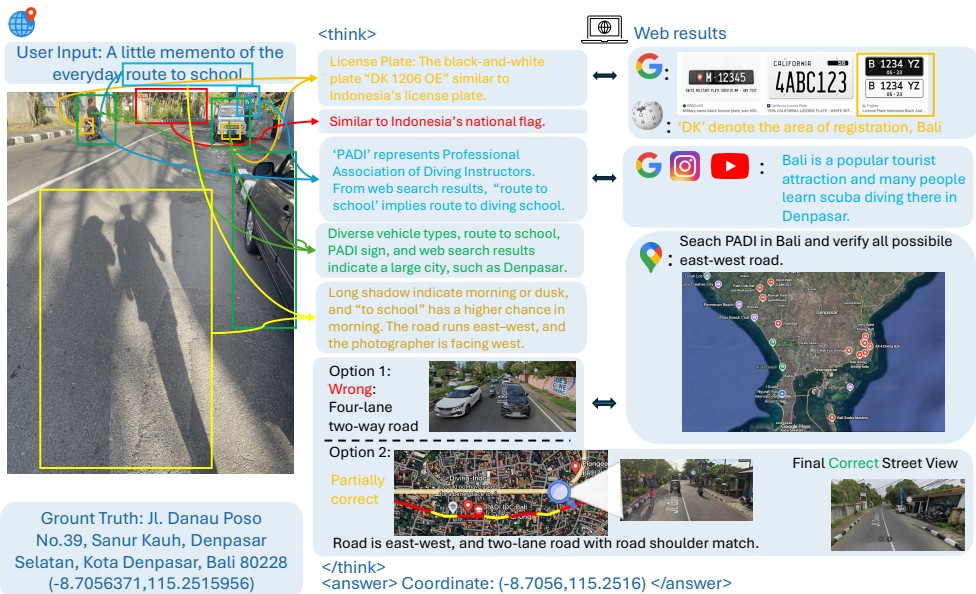

Figure 1: Illustration of a complete search and reasoning process for a `WhereBench` sample.

internet access are unavailable. Besides, they primarily report distance-threshold accuracy (Acc@X km), emphasizing outcome metrics over faithful, step-level reasoning, and rarely include human-verified annotations of the decision process.

To this end, we introduce `WhereBench`, a benchmark for web-assisted geolocation that challenges models to localize using vision-grounded reasoning and web-search tools across two scales of locations. Specifically, `WhereBench` comprises two complementary tasks: (1) `WhereCountry`, a country-level localization task with 500 curated panorama images; and (2) `WhereStreet`, a harder subtask with 310 manually verified images (188 from Bilibili[1], 122 from YouTube[2]) that asks models to identify street-level locations with reasoning and web searching. An illustration is shown in Figure 1, and a global geographic data distribution is visualized in Figure 2a.

For evaluation, `WhereBench` goes beyond outcome-only metrics. We assess both coordinate predictions and hierarchical textual localizations and explicitly consider the quality of model reasoning. Using human-annotated visual cues for answering these questions, we compute calibrated correlations between a model's reasoning traces and the final answer, where higher correlation indicates more faithful model reasoning. We also explore the use of leveraging web search for both subtasks in `WhereBench`. Overall, our `WhereBench` offers a fine-grained measurement of model reasoning fidelity and evidence use that complements the final answer metrics, yielding a clearer picture of how models think, leverage external evidence, and conclude to final answers.

We evaluate 12 leading VLMs with or without web search on our two subtasks and draw several insights from their results. Across the benchmark, we find that **closed-source models dominate**: Gemini-2.5-Pro achieves the best overall accuracy at 56.32%, while the strongest open-weight model, GLM-4.5V, lags behind at 34.71%, with most others near chance (18.50%). Drilling down by subcategory, we observe that, contrary to expectations, **neither deeper reasoning nor web search consistently improves performance on `WhereCountry`**: for instance, GPT-5 (high reasoning) drops by up to 2.5%, and GPT-4o loses 13.2% with web search. In contrast, **web access helps in `WhereStreet`**, where richer visual clues are available, yielding an average 6.5% relative boost. Together, these results highlight the challenges current VLMs face in geolocation and point to the need for more specialized capabilities beyond generic reasoning or web access.

[1] https://www.bilibili.com/
[2] https://www.youtube.com/

Table 1: Comparison of geolocation benchmarks and their properties.

| Benchmark | # Test Images | Locatability | Image Sources | Human Verified | Reasoning Process | Metrics | Web Tool Use |
|---|---|---|---|---|---|---|---|
| IM2GPS | 237 | ✗ | Flickr | ✗ | ✗ | Acc@k | ✗ |
| YFCC4k | 4,000 | ✗ | YFCC100M | ✗ | ✗ | Acc@k | ✗ |
| LLMGeo | 1,000 | ✗ | GSV | ✗ | ✗ | Acc@k | ✗ |
| GeolocationHub | 20,000 | ✗ | GSV | ✗ | ✗ | Acc@k, GeoScore | ✗ |
| Fairlocator | 1,200 | ✗ | GSV | ✗ | ✗ | City-level accuracy | ✗ |
| GPTGeoChat | 1,000 | ✗ | Shutterstock | ✓ | ✗ | Acc@k | ✗ |
| GeoChain | 2,088 | ✓ | Mapillary | ✗ | ✓ | Acc@k, Pass score | ✗ |
| WhereBench | 810 | ✓ | GSV+private | ✓ | ✓ | Acc@k, thinking score hierarchical match | ✓ |

## 2 RELATED WORK

### 2.1 VISION LANGUAGE MODELS AND AI AGENT

Vision-language models have evolved rapidly across three main paradigms: non-reasoning VLMs, reasoning-enhanced VLMs, and agentic VLMs. Non-reasoning VLMs form the foundation of multimodal AI, spanning both closed-source and open-source variants. Leading closed-source models (OpenAI, 2023; Reid et al., 2024; Hurst and many others, 2024) demonstrate strong visual understanding and language generation capabilities through direct inference without explicit reasoning steps. The open-source ecosystem (Liu et al., 2023; Wang et al., 2024a; Chen et al., 2023; Yao et al., 2025; Lu et al., 2024; Chen et al., 2024) provide accessible alternatives that often match or exceed closed-source performance on specific benchmarks. Reasoning-enhanced VLMs represent the next evolution, incorporating systematic multi-step reasoning capabilities. While closed-source reasoning models (OpenAI, 2025b;a; Anthropic, 2025) engage in extended deliberation before producing responses, the open-source community has developed corresponding reasoning models (Shen et al., 2025; Team et al., 2025; Deng et al., 2025; Xu et al., 2024; Huang et al., 2024; Chen et al., 2025) that employ chain-of-thought reasoning and self-reflection mechanisms to enhance complex visual reasoning tasks. Agentic VLMs extend beyond reasoning to incorporate tool use and environmental interaction capabilities. These models integrate with external APIs and interactive environments to solve complex real-world tasks like user interface understanding (You et al., 2024), web navigation (He et al., 2024) and reasoning tasks (Hu et al., 2024), and embodied AI tasks (Yang et al., 2024b; Zhang et al., 2024). While recent work has explored VLM geolocation capabilities (Mendes et al., 2024; Wang et al., 2024b), systematic evaluation of web-assisted geolocation remains underexplored. These developments collectively establish VLMs as versatile AI systems capable of sophisticated multimodal understanding and interaction.

### 2.2 GEOLOCATION DATASETS AND BENCHMARKS

Research on image geolocation began with retrieval-based approaches such as IM2GPS (Hays and Efros, 2008), later reframing the task as classification over geocells with PlaNet (Weyand et al., 2016). Subsequent work revisited retrieval and hybrid strategies, providing stronger baselines and standardized splits like Im2GPS3k (Vo et al., 2017), while large-scale corpora such as YFCC100M (Thomee and et al., 2016) and Google landmark datasets (Weyand et al., 2020) enabled training at global scale. Challenge series like MediaEval Placing (Choi et al., 2014) and geographically balanced sets such as GWS15k (Clark et al., 2023) further shaped evaluation protocols. Parallel to these efforts, new datasets explicitly emulate human gameplay, such as PIGEON's GeoGuessr-derived benchmark (Haas et al., 2024), enriching the evaluation of multi-view and panorama-based reasoning. With the rise of LLMs and VLMs, researchers have begun probing their geospatial knowledge (Roberts et al., 2023; Bhandari et al., 2023). Recent studies such as GeoReasoner (Li et al., 2024c) and GAEA (Campos et al., 2025) constructed large street-view based datasets for pretraining, while benchmarks such as GPTGeoChat (Mendes et al., 2024), GeoChain (Yerramilli et al., 2025), and FairLocator (Huang et al., 2025) reveal both strong geolocation capabilities and risks of privacy leakage and bias. Complementing previous works, our work proposes a multi-scale geolocation benchmark with verified human-written key clues and reasoning process assessment to probe the ability of VLMs to identify locations.

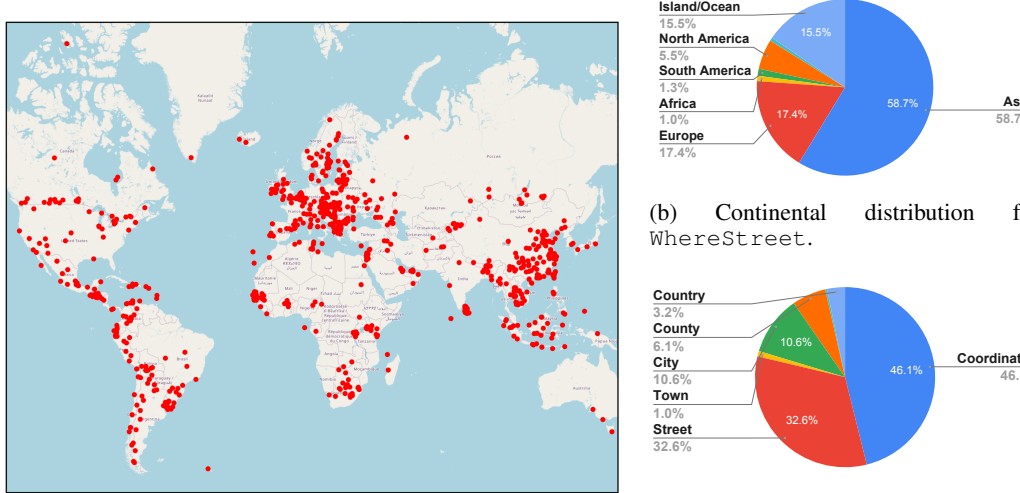

(a) All locations in `WhereBench` shown on a global map.

(b) Continental distribution for `WhereStreet`.

(c) Answer types (localization levels) for `WhereStreet` task.

Figure 2: Statistics of `WhereBench`, which reflects global coverage of geolocations (2a and 2b) at different localization levels (2c).

Table 1 shows a detailed comparison between existing geolocation benchmarks and `WhereBench`. Unlike most benchmarks which only adopt outcome-based metrics, `WhereBench` weight models' intermediate thinking ability and reflects it onto the final results, reducing answer-only restrictions while preserving global coverage. Moreover, `WhereBench` has a finer answer granularity, with both text and coordinate outputs that resemble how humans solve geolocation tasks.

## 3 WHEREBENCH

Our `WhereBench` consists of two tasks: 500 `WhereCountry` examples for coarse-grained recognition-driven country identification and 310 `WhereStreet` instances for fine-grained evidence-driven localization. To ensure fairness and robustness, we design the benchmark to achieve global coverage and balance across regions, as demonstrated in Figure 2a, showing all image coordinates in the world map. We will first dive into details about each data split, then the metrics employed for both *final answer* and *model thinking* evaluations.

### 3.1 WHERECOUNTRY

The `WhereCountry` task comprises multiple-choice question answering (MCQA) examples paired with one image, where each option corresponds to a country. Specifically for each sample, we provide a 360° panoramic image, a question asking "*Which country was this taken in?*", and four candidate countries with one correct answer. To increase the sample difficulty, we select incorrect country options from geographically adjacent countries to the target one from United Nations geoscheme[3]. Alternatively, when there are fewer than three geographically adjacent countries, we select countries that are culturally related to the target one defined in United Nations Regional Groups [4]. We start with the annotated GeoComp (Song et al., 2025) dataset and randomly sampled 8,041 images. To keep samples challenging, we utilize open-weight models to filter out simple cases, such as Street View image with national flags and unique characters in storefronts/ads, or images with limited informative clues, resulting in 680 high-quality samples. Detailed data filter process is in Appendix C. We then validate each sample's gameplay metadata in GeoComp to ensure each sample was attempted with a valid score by a real player. We rank samples by score and select the top 500 images for `WhereCountry`.

---

[3]https://en.wikipedia.org/wiki/United_Nations_geoscheme
[4]https://en.wikipedia.org/wiki/United_Nations_Regional_Groups

## 3.2 WHERESTREET

Beyond the coarse-grained country-level setting in `WhereCountry`, `WhereStreet` introduces a more challenging, fine-grained localization regime. Samples in `WhereStreet` contain more detailed visual cues that may help models pinpoint the exact location. We elaborate on the multi-scale localization levels and key clue annotation process for reasoning evaluation.

**Multi-scale Localization.** There are two answer types in `WhereStreet`: coordinate-based and text-based. Each text-based answer is classified into one of the six answer types: $AnswerType$ = [`street, town/subdistrict, city, county/district, province/state, country`]. Figure 2b summarizes continental coverage statistics, and we show each percentage of answer type for `WhereStreet` task in Figure 2c. Most `WhereStreet` items target precise localization (coordinates, street, or town), with smaller fractions at city/county and higher administrative levels.

**Key Clue Annotation for Reasoning Process Evaluation** We meticulously collect 503 publicly available English- and Chinese-language videos that document full step-by-step geolocation reasoning process. We transcribe these videos with Gemini-2.5-pro (Comanici et al., 2025) and extract candidate key clues from the transcription (see prompts in Appendix A)). We define valid key clues strictly as visual features observable in the image (*e.g.*, road markings, signage language, pole types), stripping downstream inferences so that the same feature can support different chains of reasoning. We then recruit 7 PhD student volunteers with proficient English and Chinese levels to inspect each key clue. Volunteers are required to verify text-based answers by administrative granularity as defined by $AnswerType$, and re-annotate the answer as coordinate when text alone is insufficient or ambiguous (see details in Appendix C). The inspection process yields 310 samples with 861 verified key clues, which are utilized to evaluate model thinking processes. Auxiliary "hint" information is recorded as separate metadata to contextualize difficulty without leaking answers when it is mentioned in the video and used as a supporting message to help narrow the final results (*e.g.*, "this image was taken at 5:30 pm" or "this image was taken on my way to school").

## 3.3 METRICS

**MCQA and Hierarchical Final Answer Evaluations.** We report different metrics for the two subsets. For `WhereCountry` paired with country-level MCQA, we use standard multiple-choice accuracy as the metric. For `WhereStreet` with precise coordinate, we follow previous studies (Vo et al., 2017; Weyand et al., 2016) and compute distance-based accuracy at multiple thresholds (e.g., 1 to 200 km). As for `WhereStreet` questions with street-level answers, we evaluate model predictions using a novel hierarchical path score, which reflects the granularity of correctly identified geographic attributes. Each predicted location is decomposed into a hierarchical sequence of levels: Country → Province/State → City → County → Town/Subdistrict → Street. Starting from the root (country), the model receives one point for each consecutive level that matches the ground truth.

Formally, let $\mathbf{y} = (y_1, \ldots, y_k)$ be the ground-truth locations and $\hat{\mathbf{y}} = (\hat{y}_1, \ldots, \hat{y}_k)$ the predicted locations. Then, the hierarchical path score id defined as:

$$\text{HPS}(\hat{\mathbf{y}}, \mathbf{y}) = \max\{ j \mid \hat{y}_i = y_i \ \forall i \leq j \}, \tag{1}$$

which counts the length of the longest correct prefix between the prediction and the ground truth along the location hierarchy. For example, suppose the ground truth is {A street, B county, C city, D province, China}, and the prediction is {E street, F county, C city, D province, China}. The answer type is street, and the hint is "The image is taken in China". The base is China and the target is street. Because the hint specifie China, the base is adjusted to province. From street to province, there are five hierarchical levels($k = 5$). The prediction matches at the city level but is incorrect at the street and county levels, $c = 2$. Thus, the final score is 0.4.

**Overall Performance Accuracy (%).** To combine both splits into a holistic benchmark, we compute the overall performance as the weighted-average of three components-WHERECOUNTRY (country accuracy), WHERESTREET (text answer score), and WHERESTREET (coordinate accuracy at Acc@1 km), where weights are proportional to their respective item counts:

$$\text{Overall} = \frac{N_C \, p_C + \sum_{s \in \{\text{bili,yt}\}} n_T^{(s)} s_T^{(s)} + \sum_{s \in \{\text{bili,yt}\}} n_D^{(s)} a_{1\text{km}}^{(s)}}{N_C + \sum_{s \in \{\text{bili,yt}\}} n_T^{(s)} + \sum_{s \in \{\text{bili,yt}\}} n_D^{(s)}}. \tag{2}$$

Here, $N_C = 500$ is the number of WHERECOUNTRY (country) items; $p_C \in [0, 1]$ is the corresponding accuracy. For each source split $s \in \{\text{bili}, \text{yt}\}$, $n_T^{(s)}$ is the number of WHERESTREET text-based items with mean answer score $s_T^{(s)} \in [0, 1]$, and $n_D^{(s)}$ is the number of WHERESTREET coordinate-based items with Acc@1 km $a_{1\text{km}}^{(s)} \in [0, 1]$.

**Thinking Score Evaluation.** Beyond evaluating only the final answers, we propose a novel metric to probe the internal thinking patterns, capturing a deeper sense of the model's internal behaviors. For each instance we annotate a set of $K$ key clues $\mathcal{C} = \{c_1, \ldots, c_K\}$. Given a model's reasoning trace $R$, we evaluate, for each clue $c_i$, whether it is used to narrow candidates or support the conclusion. Let $s_i \in \{0, 1\}$ indicate the decision (1 = used, 0 = not used). The vanilla thinking score is the fraction of clues that are used:

$$\text{Thinking-Score}_{\text{vanilla}} = \frac{1}{K} \sum_{i=1}^{K} s_i .  \tag{3}$$

To make the thinking score more robust and better reflect true reasoning ability, we reweight key clues by their marginal contribution to narrowing the candidate location, as certain clues contribute more to identifying the location than others. In detail, we estimate clue importance using Shapley values (Rozemberczki et al., 2022), so that the reasoning score is tied more closely to how much each clue actually helps in reducing uncertainty. Formally, let $C$ denote the set of key clues for an instance. Define a value function $v : 2^C \to [0, 1]$, where for any subset $S \subseteq C$, $v(S)$ is the expected answer quality if the model only has access to clues in $S$. Then for each clue $i \in C$, the Shapley weight $w_i$ is defined by:

$$w_i = \sum_{S \subseteq C \setminus \{i\}} \frac{|S|! \, (|C| - |S| - 1)!}{|C|!} \left( v(S \cup \{i\}) - v(S) \right), \quad \sum_{i \in C} w_i = v(C) .  \tag{4}$$

We implement $v(S)$ by enumerating all $2^{|C|}$ subsets $S$, prompting the judge (Gemini-2.5-Pro) to assign the achievable answer quality using only clues in $S$. From those values, we compute the full Shapley vector $\{w_i\}$ and compute the reweighted thinking score as

$$\text{Thinking-Score}_{\text{reweighted}} = \sum_{i \in C} w_i \cdot s_i  \tag{5}$$

where $s_i \in \{0, 1\}$ is the binary credit for clue $i$, indicating that the model correctly identified clue $i$ in its reasoning. Note that the Shapley weights are computed once per sample under a fixed judge and prompt and then reused for all evaluated models, ensuring an efficient and reproducible Thinking-Score across evaluations. In Section 4.3, we showcase that the reweighted Thinking-Score has an average 0.03 higher correlation than the vanilla version with the final answer, which justifies its use.

## 4 EXPERIMENT

We first present detailed descriptions of our experimental settings, followed by an overview of model performance on WhereBench. Then, to probe different model capabilities across geolocation scales and tasks, we examine the separate splits of the dataset in depth.

**Experimental Setup.** We evaluate diverse open-weight and closed-source models which are categorized as follows: (i) **Open-weight VLMs.** Baseline VLM models such as Qwen-2.5-7B (Yang et al., 2024a). (ii) **Open-weight VLMs with built-in tool use.** Recent open-weight models expose native tool abilities (e.g., zoom/resize). We include GLM-4.5V (Team et al., 2025), DeepEyes-7B (Zheng et al., 2025), and Skywork-R1V3 (Shen et al., 2025). (iii) **Closed-source VLMs.** We evaluate Claude4-Opus (Anthropic, 2025) and Claude4-Sonnet (Anthropic, 2025) as strong closed baselines. (iv) **Closed-source VLMs with web search.** Many VLMs support web-enabled retrieval. We evalute both reasoning-enabled and standard variants, including Gemini-2.5-pro (Comanici et al., 2025), Gemini-2.5-flash (Comanici et al., 2025), GPT4o (Hurst and many others, 2024), o3 (OpenAI, 2025b), o4-mini (OpenAI, 2025b), and GPT5 (OpenAI, 2025a). We also report results with web search disabled for each model.

Our evaluation is guided by two prevailing hypotheses regarding VLM scaling. The reasoning hypothesis: that increasing reasoning depth allows models to better synthesize conflicting visual clues,

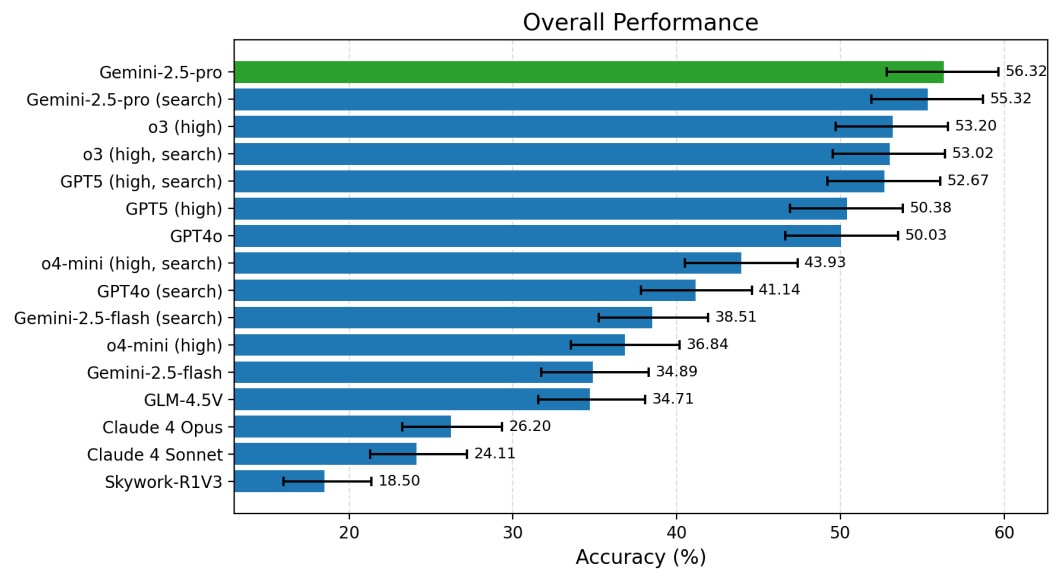

Figure 3: Overall performance combining both the `WhereCountry` and `WhereStreet` results.

leading to higher geolocation accuracy. The retrieval hypothesis: that access to external knowledge via web search improves performance by allowing models to verify visual landmarks. We follow all official or recommended inference settings for each VLM and use the native web APIs for internet access. Textual evaluation for `WhereStreet` follows an LLM-as-a-Judge protocol with Gemini-2.5-pro with an average Kappa agreement with human judges exceeding 0.75 (Appendix B). The complete prompts for querying VLMs and evaluations are in Appendix A.

**Overall performance.** Figure 3 ranks VLMs by the micro-averaged accuracy across `WhereBench`. We can conclude several ideas from the figure: 1. Closed-source models lead the performance, *Gemini-2.5-pro* attains the highest overall score, with its web-enabled variant close behind; reasoning-focused baselines (*o3* (high) and *GPT5* (high)) form the next tier with narrow gaps. 2. Claude models underperform relative to other closed models, while the top open-weight VLMs, such as GLM-4.5V, are competitive with several lightweight closed-source baselines, indicating a narrow but existing gap between closed and open-weight models. 3. Counterintuitively, applying web search does not always ensure a superior performance; it varies depending on indicative visual clues, as further analyzed in Sec 4.2.

Notably, compared with performance reported under more constrained geolocation setups such as geocell classification against a fixed database or geographically limited city-level answer (Huang et al., 2025; Li et al., 2025; Vo et al., 2017), VLMs generally attain lower absolute scores on `WhereBench`. This gap underscores that `WhereBench` imposes different—and harder—requirements: broader geographic coverage, cross-source variability, and mixed evaluation targets, where search and long-form reasoning are not guaranteed advantages but must be *selective* and *precise*. We further provide ablations regarding tool using, reasoning effort, input visual clues in the following sections.

## 4.1 WHERECOUNTRY

Figure 4 summarizes models' country–level accuracies on WHERECOUNTRY, from which we obtain two insights below.

**Closed models dominate, the best open model narrows but does not close the gap.** Without web access, Gemini-2.5-pro attains the highest accuracy at 68.4%, followed by o3 with a high reasoning effort. Among open-weight models, GLM-4.5V is strongest at 43.8%, whereas the remaining open-weight baselines perform around chance with an average accuracy of only 19.57%, underscoring a persistent capacity gap on geolocation tasks to proprietary models.

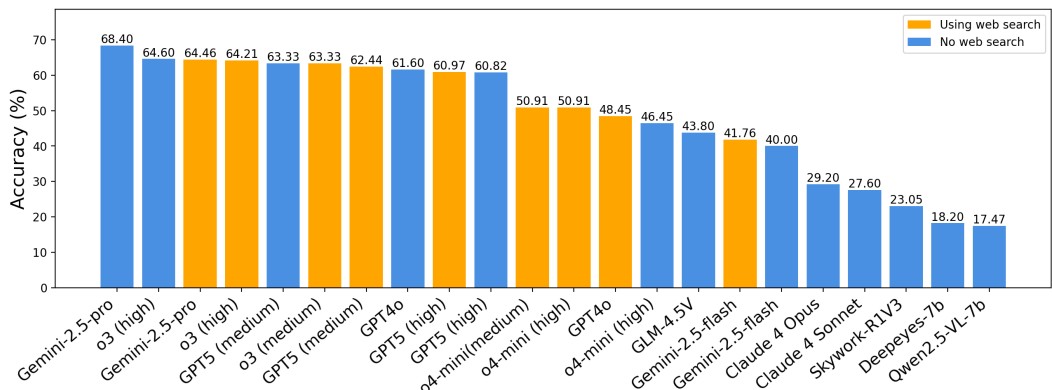

Figure 4: **Main results on `WhereCountry` ranked by accuracy.** Closed-source models lead by a large margin. Neither web search nor deeper reasoning consistently improves performance.

Table 2: Results on `WhereStreet` sourced from Bilibili and Youtube, with models as columns and different metrics as rows. Darker green indicates better results within each row.

| Models | Gemini-2.5 pro | | Gemini-2.5 flash | | o3 (high) | | o4-mini (high) | | GPT5 (high) | | GPT4-o | | Claude4 Sonnet | Claude4 Opus | Skywork R1V3 | GLM 4.5V |
|---|---|---|---|---|---|---|---|---|---|---|---|---|---|---|---|---|
| *Web* | ✗ | ✓ | ✗ | ✓ | ✗ | ✓ | ✗ | ✓ | ✗ | ✓ | ✗ | ✓ | ✗ | ✗ | ✗ | ✗ |
| *Bilibili* | | | | | | | | | | | | | | | | |
| Acc@1km | 2.13 | 6.38 | 0.00 | 2.13 | 2.13 | 2.13 | 2.13 | 2.33 | 4.26 | 2.17 | 0.00 | 0.00 | 2.22 | 2.17 | 0.00 | 2.13 |
| Acc@5km | 23.40 | 17.02 | 10.64 | 14.89 | 17.02 | 21.28 | 10.64 | 13.95 | 19.15 | 21.74 | 10.87 | 8.51 | 6.67 | 8.70 | 2.13 | 8.51 |
| Acc@20km | 40.43 | 34.04 | 29.79 | 25.53 | 34.04 | 34.04 | 21.28 | 25.58 | 34.04 | 30.43 | 26.09 | 29.79 | 22.22 | 21.74 | 17.02 | 23.40 |
| Acc@200km | 53.19 | 55.32 | 55.32 | 48.94 | 48.94 | 51.06 | 44.68 | 44.19 | 48.94 | 58.70 | 52.17 | 55.32 | 44.44 | 47.83 | 53.19 | 51.06 |
| Thinking Score | 0.436 | 0.483 | 0.351 | 0.272 | 0.425 | 0.414 | 0.401 | 0.340 | 0.249 | 0.275 | 0.273 | 0.204 | 0.149 | 0.232 | 0.192 | 0.268 |
| *YouTube* | | | | | | | | | | | | | | | | |
| Acc@1km | 58.06 | 65.63 | 46.88 | 57.29 | 54.74 | 55.21 | 27.08 | 52.69 | 50.53 | 63.54 | 46.32 | 47.37 | 29.35 | 39.33 | 7.29 | 18.95 |
| Acc@5km | 73.12 | 73.96 | 63.54 | 68.75 | 70.53 | 66.67 | 44.79 | 56.99 | 68.42 | 72.92 | 64.21 | 63.16 | 43.48 | 49.44 | 15.63 | 36.84 |
| Acc@20km | 77.42 | 80.21 | 72.92 | 70.83 | 73.68 | 71.88 | 55.21 | 63.44 | 72.63 | 76.04 | 72.63 | 70.53 | 52.17 | 56.18 | 21.88 | 53.68 |
| Acc@200km | 86.02 | 85.42 | 86.46 | 81.25 | 84.21 | 73.96 | 68.75 | 70.97 | 81.05 | 81.25 | 82.11 | 81.05 | 68.48 | 70.79 | 43.75 | 70.53 |
| Thinking Score | 0.814 | 0.803 | 0.684 | 0.665 | 0.686 | 0.789 | 0.652 | 0.572 | 0.521 | 0.354 | 0.630 | 0.492 | 0.491 | 0.540 | 0.495 | 0.609 |

**Additional effort on reasoning or web search does NOT guarantee improved performance.** To examine the impact of advanced model reasoning abilities on `WhereCountry`, we conduct controlled experiments that vary reasoning depth and web search usage.

- Increasing reasoning from medium to high yields only marginal gains: OpenAI systems achieve an average −1.03% decrease with web search, and the strong reasoning model o3 (high) improves by just 1.3%. Similarly, o4-mini (high, search) shows no improvement, while GPT-5 (high) drops by 1.47% and 2.51% with and without search, respectively. These results suggest that `WhereCountry` is perception-centric, where additional reasoning efforts can not lead to higher accuracy without precise perception.

- Web search, while offering external and real-time information, surprisingly provides *little to no benefit* with an average of 1.72% drop. In fact, GPT-4o suffers a substantial 13.2% drop when web search is enabled. These drops reflect both a text-only bottleneck in query formulation and potential limitations of search engines for geolocalization. `WhereBench` thus exposes a system-level failure to turn internet access into geolocation gains, motivating tighter integration between visual understanding and web tools.

Together, these findings demonstrate that neither deeper reasoning nor web search consistently improves performance on `WhereCountry`. Instead, they underscore the challenging nature of the benchmark and the need for better visual-search integration to support localization with limited visual clues. A detailed case study is provided in Section 4.4.

Table 3: Answer and thinking scores on `WhereStreet` sourced from Bilibili and Youtube, with models as columns and different metrics as rows.

| **Models** | Gemini-2.5 pro | | Gemini-2.5 flash | | o3 (high) | | o4-mini (high) | | GPT5 (high) | | GPT4-o | | Claude4 Sonnet | Claude4 Opus | Skywork R1V3 | GLM 4.5V |
|---|---|---|---|---|---|---|---|---|---|---|---|---|---|---|---|---|
| *Web* | ✗ | ✓ | ✗ | ✓ | ✗ | ✓ | ✗ | ✓ | ✗ | ✓ | ✗ | ✓ | ✗ | ✗ | ✗ | ✗ |
| | | | | | | | | *Bilibili* | | | | | | | | |
| **Answer Score (%)** | 26.1 | 26.8 | 15.3 | 20.1 | 23.9 | 22.0 | 16.5 | 20.8 | 23.6 | **28.1** | 23.2 | 19.2 | 12.7 | 10.6 | 13.4 | 19.6 |
| Thinking Score | 0.520 | 0.459 | 0.418 | 0.370 | 0.481 | 0.548 | 0.382 | 0.347 | 0.375 | 0.310 | 0.325 | 0.232 | 0.210 | 0.223 | 0.197 | 0.317 |
| | | | | | | | | *YouTube* | | | | | | | | |
| **Answer Score (%)** | 79.6 | 84.7 | 61.6 | 72.4 | 79.7 | **90.1** | 61.2 | 67.4 | 78.9 | 75.6 | 71.9 | 71.0 | 38.3 | 50.8 | 33.2 | 56.8 |
| Thinking Score | 0.762 | 0.742 | 0.636 | 0.644 | 0.646 | 0.675 | 0.644 | 0.606 | 0.499 | 0.315 | 0.685 | 0.509 | 0.468 | 0.522 | 0.511 | 0.663 |

| **Models** | **o3** | | | **o4-mini** | | | **GPT5** | | |
|---|---|---|---|---|---|---|---|---|---|
| Reasoning | Low | Med. | High | Low | Med. | High | Low | Med. | High |
| | | | | | *Bilibili* | | | | |
| **Answer (%)** | 23.5 | **26.8** | 22.0 | 15.2 | 19.8 | **20.8** | 25.4 | 26.5 | **28.1** |
| Thinking | 0.46 | 0.50 | 0.55 | 0.38 | 0.38 | 0.35 | 0.09 | 0.23 | 0.31 |
| | | | | | *YouTube* | | | | |
| **Answer (%)** | 77.2 | 79.7 | **90.1** | 63.6 | **72.9** | 67.4 | 81.9 | **83.1** | 75.6 |
| Thinking | 0.70 | 0.59 | 0.68 | 0.74 | 0.63 | 0.61 | 0.18 | 0.22 | 0.32 |

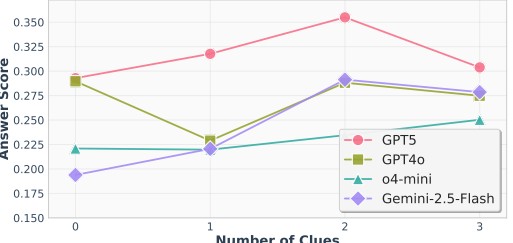

(a) Ablation on reasoning effort with web search on `WhereStreet`.

(b) Effect of number of human-annotated key clues as extra context.

Figure 5: Ablations on reasoning effort (left) and number of human-annotated key clues (right) on `WhereStreet`.

## 4.2 WHERESTREET

The main results for `WhereStreet` are shown in Table 2 for coordinate-based answers and Table 3 for questions paired with street-level text answers. We partition the data by source (Bilibili: 188 samples; YouTube: 122 samples). Overall, for coordinates, Gemini-2.5-pro with web achieves the highest Acc@1km: 6.4% (Bilibili) and 65.6% (YouTube). For text, GPT5 (high reasoning, web) yields the best Bilibili answer score (0.28), while o3 (high reasoning, web) leads on YouTube (0.90). We provide complete results in Appendix D and detailed case studies in Appendix E.2.

**Web search helps when facing more detailed visual clues.** In Table 3, web access improves the ability of models to identify street-level locations given the image, where the image generally contains more fine-grained visual details that enable audience to infer street-level answer. This is evidenced by an averaged relative boosts of both 6.5% on two data sources (*e.g.*, 21.4 *vs.* 22.8 on Bilibili and 72.2 *vs.* 76.9 on YouTube). Moreover, GPT5 gains substantially with web access on the Bilibili data source — moving from below Gemini-2.5-Pro in the no-web condition to among the top models with web enabled (*e.g.*, GPT5: 28.1 *vs.* Gemini-2.5-pro: 26.8).

**`WhereStreet` with more visual details requires certain level of reasoning.** Figure 5a reports results for o3, o4-mini, and GPT-5 across three reasoning effort levels with web search enabled. These models show consistent gains when moving from low to medium effort—an average relative improvement of 14.0% on Bilibili and 5.9% on YouTube (*e.g.*, 21.4 *vs.* 24.4 on Bilibili and 74.2 *vs.*. 78.6 on YouTube). However, increasing the effort further brings no additional benefit (*i.e.*, medium 51.5 *vs.* high 50.7 on average across both sources and all models). This suggests that while a moderate level of reasoning is helpful for interpreting the richer visual details in `WhereStreet`, excessive reasoning offers decreased returns. In other words, reasoning aids comprehension but is not the ultimate solution for fine-grained geolocation, where precise recognition and grounding remain the primary challenges. We present complete results in Appendix D, where coordinate-based scenarios also shows a similar trend.

## 4.3 ABLATION STUDY

To justify the use of the proposed reweighted Thinking-Score and human-annotated key clues, we conduct ablation studies on `WhereStreet` and give the following findings.

Table 4: Pearson correlations across models between answer and (i) reweighted thinking score (Our metric) and (ii) thinking score.

| | Gemini-2.5-pro | Gemini-2.5-flash | o3 (high) | o4-mini (high) | GPT5 (high) | GPT4-o | Claude4-Sonnet | Claude4-Opus |
|---|---|---|---|---|---|---|---|---|
| Reweighted Pearson | 0.248 | 0.227 | 0.221 | 0.229 | 0.133 | 0.389 | 0.305 | 0.345 |
| w/o reweight | 0.236 (-0.012) | 0.182 (-0.045) | 0.143 (-0.078) | 0.251 (+0.022) | 0.078 (-0.055) | 0.323 (-0.066) | 0.336 (+0.031) | 0.336 (-0.009) |
| | Gemini-2.5-pro (search) | Gemini-2.5-flash (search) | o3 (high, search) | o4-mini (high, search) | GPT5 (high, search) | GPT4-o (search) | Skywork-R1V3 | GLM-4.5V |
| Reweighted Pearson | 0.246 | 0.209 | 0.219 | 0.289 | 0.118 | 0.316 | 0.208 | 0.283 |
| w/o reweight | 0.176 (-0.070) | 0.149 (-0.060) | 0.165 (-0.054) | 0.275 (-0.014) | 0.055 (-0.063) | 0.281 (-0.035) | 0.203 (-0.005) | 0.314 (+0.031) |

**Model thinking scores indicate the answer quality and reweighting tightens it.** To prove the effectiveness of the proposed thinking evaluation, we compute Pearson correlations between answer score and (i) the raw thinking score and (ii) the *reweighted* thinking score (Sec. 3.3); results appear in Table 4. Reweighting strengthens the correlation with an average 13.70% higher, aligning with our goal of assessing process quality rather than only final correctness. Qualitative analysis shows that models frequently ground several cues correctly yet miss a decisive clue, yielding incorrect predictions. We specifically examined GPT-5 to understand its low correlation and found that its outputs are high-level summaries rather than complete reasoning traces, consistent with GPT-5's limited disclosure of detailed thinking steps for intellectual-property and safety reasons.

**Human-verified clues are accurate, providing more clues as input generally yields higher scores.** To validate the utility of our annotated key clues, we designed an experiment to randomly select 1, 2, or 3 clues from the annotated key clues list and prepend them as context with the question and evaluate whether models can gain extra score. We evaluate textual-based samples on GPT4o, o4-mini, GPT5, and Gemini-2.5-Flash without web access, and the results are shown in Figure 5b. The answer score increases with more clues. We attribute the answer score fluctuation to the difference in each clue's true value and GPT4o's performance drop to the base model's limited capability.

## 4.4 Case Study

We provide a few typical VLM failure reasons: (1) Failure to utilize visual clues for narrowing down exact locations. In Appendix E.1, GPT-4o with web search overlooked tree types and fencing style in the background, concluding on a wrong final answer. Whereas without web searching let GPT-4o capture the details, leading to the correct answer. (2) Overthinking. Appendix E.2 shows an example that models could overthink and contradict to themselves. GLM-4.5-V successfully inferred the territory and coastline structure, but rejected its correct assumption with a self-contradictory reason. This might be due to lengthy thinking process containing unnecessary aha moments (Guo et al., 2025), making models stuck in hesitancy. (3) Incomplete searching. Appendix E.2 shows another example of Gemini-2.5-pro with web search. Gemini-2.5-pro correctly identified the key visual elements and projected reasonable assumptions. Yet, constrained by current tool-use capabilities (*e.g.* suboptimal search queries, limited search iterations, or restricted retrieval context length), the answering process was terminated early and the model failed to locate the final coordinates. Beyond these qualitative instances, we conduct a systematic quantitative error analysis in Appendix D.

## 5 Conclusion

We introduce `WhereBench`, a standardized benchmark for image geolocation across both country and street scale. Designed for balance, verifiability, and global coverage, `WhereBench` unifies two complementary tasks: `WhereCountry` (recognition-centric) and `WhereStreet` (analysis-and-evidence) to deliver multi-granularity, multi-level assessment. Beyond coordinate accuracy and hierarchical textual localization, we contribute a process-aware protocol: an LLM-as-a-Judge rubric that verifies whether key visual clues are actually used, together with a Shapley-reweighted thinking score that attributes credit by marginal contribution. Extensive experiments reveals that strong closed models excel on `WhereCountry` without retrieval, while search aids `WhereStreet` with model- and distribution-dependent gains. Overall, `WhereBench` is challenging, and state-of-the-art VLMs remain below human-level precision in fine-grained localization. We aim for `WhereBench` to serve as a clear target with standardized protocols that facilitates fair comparison, drive sustained progress, and clarify how VLMs and agents reason with images and leverage web evidence.

ETHICS STATEMENT

`WhereBench` is developed to probe the geolocation capabilities of vision–language models and not to facilitate privacy invasion or surveillance. Nonetheless, image geolocation poses clear privacy and misuse risks (*e.g.* , stalking, targeted harassment, illicit tracking, or other abusive surveillance). To mitigate these risks during dataset curation we only collected publicly available items that (i) contain an explicit final location reveal, (ii) are non-synthetic, and (iii) do not contain personally identifying information; items failing these criteria were excluded. For each retained sample we extract a single canonical frame and explicitly remove EXIF and auxiliary metadata; candidate visual clues were restricted to verifiable visual features (e.g., road markings, signage styles, vegetation) and screened by trained annotators (see Appendix B). Our intent in releasing `WhereBench` is to support research-focused evaluation of model capabilities rather than to enable applied geolocation systems. According to this intent, any public release will include clear usage terms and guidance that discourage malicious applications (*e.g.* , recommending access only to vetted researchers, providing redacted versions where appropriate, and documenting responsible use). Finally, we emphasize directions for future work to reduce risk: developing model refusal policies and classifier guidance that teach models when to decline fine-grained location requests, and adding audit trails for retrieval-enabled evaluations so that downstream misuse is harder to automate.

REPRODUCIBILITY STATEMENT

We provide detailed dataset construction steps (Appendix C), prompt templates and evaluation protocols (Appendix A), and full experimental results and ablations (Appendix D and E). All model settings are specified in Section 4. Supplementary materials include the `WhereBench` image list, key-clue annotations, evaluation scripts, and cached web queries. Together, these resources ensure that construction of `WhereBench` and its findings can be reliably reproduced.

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

# Appendices

## A PROMPTS

We present the full prompts used for LLM-as-a-Judge. Table 5 shows the prompt for evaluating the answer score when the output is text. Table 6 is the prompt used to check whether key clues appears in the model's response, resulting in the vanilla thinking score. In Table 7, it is the complete prompt for computing the Shapley value of each clue. Finally, Table 8 shows the prompt used to extract key clues from transcripts produced by Gemini-2.5-pro.

## B VALIDATING LLM-AS-A-JUDGE SETTING

We validate the reliability of our LLM-judge (Gemini-2.5-pro) by computing Cohen's $\kappa$ against human annotations on held-out subsets on three models: GLM-4.5-V ($n = 47$, $\kappa = 0.74$), o3 ($n = 45$, $\kappa = 0.83$), and o4-mini ($n = 59$, $\kappa = 0.70$). These values indicate strong human-model agreement, supporting the use of Gemini-2.5-pro as a reliable judge of model outputs.

## C DATA CURATION

### C.1 WHERECOUNTRY

After randomly sampled 8,041 images, we utilize Qwen-2.5VL-7B (Wang et al., 2024a) to filter out simple and direct cases such as Street View images with national flags, unique characters or letters in the storefronts/ads, car plates, etc, resulting in 2,359 images. Then, we apply a second filter, LLaVA-OneVision (Li et al., 2024b), to flag residual low-quality cases where images may not contain enough information to pinpoint the exact country, leaving 680 high-quality samples. Failed image samples are shown in Figure 6.

### C.2 WHERESTREET

We curate public social-media channels[5] that regularly publish image/video geolocation challenges with an explicit final reveal. We apply the following criteria: (i) content is publicly accessible; (ii) each item contains (or links to) a definitive location; (iii) footage appears non-synthetic; (iv) no personally identifying information. Items failing these criteria are excluded. For each selected video, we generate an ASR transcript using Gemini-2.5-pro. Given the raw transcript, Gemini-2.5-pro proposes a set of candidate key clues: short sentences that plausibly reference visual evidence (e.g., "left-hand traffic," "blue street name plates," "Andean highlands vegetation"). 7 trained annotators review each item after watching the original video. Annotators independently write the final answer as revealed by the video. If the final textual answer cannot faithfully represent the final answer, annotators utilize Google Maps to manually cross-check and verify the final location and note the exact coordinate.

For every LLM-proposed clue, annotators check against the raw video. We keep cues that can be verified visually (landforms, road markings, language script without specific place names, license-plate format, vegetation, architecture) and remove subjective indications where models might conduct diverse deductions. Annotators label the finest administrative level that is correctly mentioned by the video. Additionally, any external information that is used by the video but absent from the image is saved for inference. Annotators capture the input image as a single canonical frame from the original video, excluding any EXIF/auxiliary metadata. To ensure annotation consistency and quality, we employed a hierarchical verification protocol. Ambiguities encountered during the annotate process were adjudicated by a lead annotator with extensive experience in geolocation. Following the initial annotation, the lead expert then verify ten samples per annotator to spot-check the quality. When a single issue was detected, a comprehensive review of that annotator's entire batch was triggered to ensure accuracy.

---

[5]https://space.bilibili.com/1078123935,https://space.bilibili.com/
1078123935,https://www.youtube.com/playlist?list=PL_japiE6QKWqMVC3JbyONau_
0CZlDTU5f,https://www.youtube.com/@GeoPeter,https://www.youtube.com/
@Nattic,https://www.youtube.com/watch?v=rl2Q9xH8e7M

Table 5: Prompt for scoring textual geolocation answers via hierarchical matched-prefix credit.

**ROLE**
You are a strict geolocation evaluator. Compare a predicted location to a ground-truth location and return *one* accuracy score as a **float** in [0.0, 1.0].

**INPUTS**
– Predicted Location: "{predicted}"
– Ground Truth Location: "{ground_truth}"
– Granularity to Judge (`answer_type`): "{answer_type}"    (one of: country | province/state | county/district | city | town/subdistrict | street)
– Hint (reference only; do not copy): "{hint}"

**RULES**
*1) Normalize & Parse*
– Case/diacritic-insensitive; ignore punctuation/extra whitespace; accept common aliases (e.g., "NYC"="New York City", "München"="Munich").
– Use this ordered hierarchy (down→top): **street > town/subdistrict > city > county/district > province/state > country**.
– Map obviously equivalent administrative terms across countries (e.g., borough/parish/district). Do not invent missing components.

*2) Define the* SCORING PATH *(denominator)*
– Let $L_{target}$ be the level named by {answer_type}.
– Determine a base level $L_{base}$:
• If the Hint names a level $L_{hint}$ that is *consistent with* the Ground Truth, set $L_{base} =$ one level *below* $L_{hint}$ (treat the Hint as free information; exclude it from credit).
• Otherwise (no usable Hint), set $L_{base} =$ country.
– The scoring path is the contiguous list of levels from $L_{base}$ (inclusive) up to $L_{target}$ (inclusive).
Denominator $= k =$ number of levels in this path ($k \geq 1$).

*3) Compute Matched Prefix Count (numerator)*
– Walk the path from $L_{base}$ downward. Count how many consecutive levels match the Ground Truth before the first mismatch.
– A level "matches" if either:
• The Predicted explicitly names the same unit as the Ground Truth at that level, *or*
• The Predicted *omits* that level but correctly names any *finer (lower)* level under the same Ground Truth parent (implicit parent credit), with no contradicting tokens.
– If the first level on the path ($L_{base}$) is wrong, matched count $= 0$.

*4) Score*
– Score $=$ matched_count/denominator $\in [0, 1]$.
– Examples when {answer_type}=`street` and Hint gives a province (e.g., "Guangdong"):
Correct city→county→town→street: $4/4 = \mathbf{1.0}$
Correct city→county→town, wrong/missing street: $3/4 = \mathbf{0.75}$
Correct city→county, wrong/missing town: $2/4 = \mathbf{0.50}$
Correct city only, wrong/missing county: $1/4 = \mathbf{0.25}$
Wrong city: $0/4 = \mathbf{0.00}$

*5) Anti-Cheating*
– If the Predicted string copies the Hint (or is trivially derived from it) *without adding any level at or below* {answer_type}, set score to **0.00**.
– **Exception:** If the Hint provides multiple-choice style constraints (e.g., "The image is in one of: UK/Canada/USA/Mexico."), do not penalize merely repeating the hinted country.

**OUTPUT (strict)**
Return only the float ($\leq 3$ decimals) inside this tag: `<answer>SCORE</answer>`

**Illustrative Examples**
*1.* GT: Beicheng Street, Zaoyang county, Xiangyang city, Hubei, China.
Pred: Niushou Town, Xiangyang city, Hubei, China.
answer-type: `street`; hint: China.
Path: street→town/sub→county→city→province ($k = 5$).
Match: county mismatches, but city matches $\Rightarrow 2$.
Score: $2/5 = 0.4$.

Table 6: Prompt used for LLM evaluation of whether a key clue was used in reasoning.

You are an expert evaluator of logical reasoning and evidence utilization.
**TASK**
Decide whether the Key Clue was actually USED within the Reasoning Process to advance or support the location inference.
**INPUT**
Key Clue: "{key_clue}"
Reasoning Process: "{thinking_process}"
**DEFINITIONS**

- **Mentioned:** the clue (or a clear synonym) is referenced in the reasoning.
- **Used:** the reasoning relies on the clue to narrow candidates, eliminate options, strengthen a hypothesis, or justify the final conclusion.
- **Dismissed:** the clue is mentioned but explicitly rejected or not carried forward.
- **Misused:** the clue is cited but interpreted incorrectly.

**ALLOWED EVIDENCE**
Judge *only* from the provided Reasoning Process. Do *not* add facts from outside knowledge or the image itself. Do *not* judge whether the final answer is correct—only whether the clue was used.
**DECISION RULES**
Answer "Yes" *ONLY* if *all* are true:

1. The clue (or a clear synonym/phrase) is mentioned or unmistakably referred to, *and*
2. The reasoning uses it to narrow, rule out, weigh options, or support the conclusion (an explicit causal link or justification).

Otherwise answer "No", including these cases:

- Mentioned as a guess, observation, or side note without narrowing/supporting.
- Mentioned then dismissed or ignored.
- Not mentioned at all (directly or via clear synonym).
- Misunderstood or misused as evidence.
- Ambiguous/uncertain whether it aided reasoning.

**OUTPUT INSTRUCTIONS**
Return:
```
<answer>Yes/No</answer>
<explanation>One brief sentence justifying the decision.
</explanation>
```
**CONSTRAINTS**

- Base your decision strictly on the Reasoning Process text above.
- If in doubt, answer "No".
- Keep the explanation to 1–2 sentences.

# D    MORE RESULTS

Here we present the complete results of WHERESTREET for textual-based answer (Table 10), coordinate-based (Table 11), and the ablation study results on reasoning effort and web search in Table 11.

**Error Analysis**    To provide a quantitative understanding of model limitations, we developed a taxonomy of three primary error types. We utilized an LLM to evaluate the reasoning traces of incorrect responses and classify them based on the following criteria. Note that these categories are not mutually exclusive. Thus, the summation of these error types could exceed 100%.

Table 7: Prompt for computing Shapley values of key clues based on their contribution to final answer quality.

---

**System:**
You are an expert in calculating Shapley values for feature attribution in machine learning models. Your task is to analyze reasoning files and calculate Shapley values for key clues based on their contribution to the final answer quality.

Follow these guidelines: 1. From initial key_clues with index, find out all the combinations. 2. For each combination of the clue, based on the Ground Truth answer and the hint, determine an anchor of which level of answer a model would finally generate. Answer-types are: Country | Province or State | county/district | city | town/subdistrict | street. Refer to the reasoning file to determine the anchor. 3. Finetune the score using the reasoning file as the gold standard; determine the exact score for each combination. 4. Similar to how the Shapley value is calculated: calculate the Shapley value for each clue. For the combination of no clues, the Shapley value is 0. For the combination of all clues, the Shapley value is 1. Each Shapley value is a float between 0 and 1.

**User:**
Here is the reasoning file content: {reasoning}

The key clues are: {gt_key_clues}. The ground truth answer is: {gt_answer}. The hint is: {hint}.

Note: Hint is supplemental information to the image; it is not a clue. Return a list of Shapley values for each clue in this format:
`<answer>[shapley_value_1, shapley_value_2, ...]</answer>`

---

Table 8: Prompt for extracting key clues from the input transcript.

---

Here is the text thinking process of how to deduce the exact location from the input image: `{text_content}`
Ignore the caption and watermark. Based on the thinking process and input image, create a comprehensive list of key steps.
Do not include any clues that are not mentioned in the text description.
Do not repeat clues.
Merge two clues if they are very similar.
Focus on the most important clues that can help deduce the location.
Format your response as a numbered list where each line starts with a number followed by a period and space (e.g., "1. The first clue."). Each key clue should be concise and accurate.

---

- Missed Visual Clues: The model fails to perceive or correctly ground decisive image-based evidence. Typical patterns include ignoring text, domain suffixes, or landmarks; contradicting clear visual signals (e.g., misidentifying language scripts); or remaining vague despite the presence of highly specific localized features.

- Overthinking: The model reaches a plausible hypothesis but discards it due to reasoning instability. Indicators include unnecessarily speculative chains of thought, unresolved contradictions (e.g., conflicting driving sides), or excessive elaboration that degrades the final answer quality compared to earlier intermediate steps.

- Incomplete Search: The model attempts to use external tools but fails in execution. Common failures include generating generic queries based on misinterpreted clues, hallucinating confirmation from irrelevant search results, or terminating the search process prematurely while uncertainties remain.

As results shown in Table 12, Gemini-2.5-pro and o3 (high) exhibit strong visual grounding, missing the fewest visual clues (35.16% and 43.04% respectively). However, they are hampered by search alignment errors ( 38–55%), where the model correctly identifies visual features but fails to translate them into effective search queries. Conversely, open-weight models such as Skywork-R1V3 and

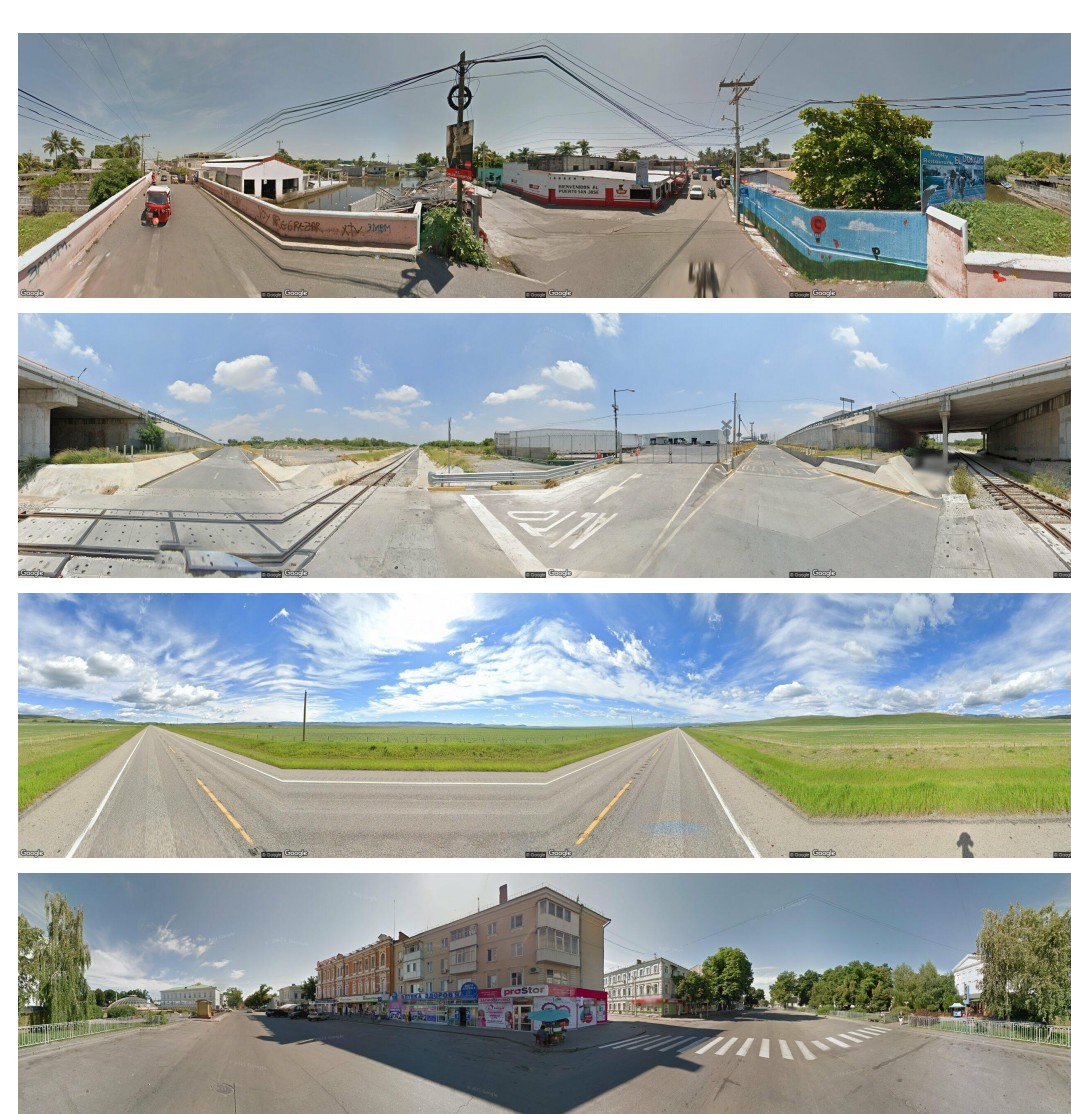

Figure 6: Failed image samples. They either have direct text to indicate the country or process relatively limited visual informative clues.

GLM-4.5V suffer from a high rate of "Overthinking" ($\approx 35\%$), where the model generates excessive reasoning steps that drift away from the ground truth constraints. This suggests that while these models have strong raw generation capabilities, they lack the reasoning stability of close-source counterparts.

# E    CASE STUDY

To better understand VLMs performance, we provide a detailed case study for `WhereCountry` and `WhereStreet`.

## E.1    WHERECOUNTRY

We present a GPT4o case study in Table 13 where GPT4o utilizes its internal knowledge, leading to the correct final answer, but a wrong answer when accessing the web.

Table 9: Geolocation accuracy by model.

| Source | Model | Samples | Acc@1km | Acc@5km | Acc@10km | Acc@20km | Acc@50km | Acc@100km | Acc@200km | Thinking Score |
|---|---|---|---|---|---|---|---|---|---|---|
| | gemini-2-5-pro | 47 | 2.13% | 23.40% | 27.66% | 40.43% | 46.81% | 48.94% | 53.19% | 0.436 |
| | gemini-2-5-pro (search) | 47 | 6.38% | 17.02% | 23.40% | 34.04% | 42.55% | 44.68% | 55.32% | 0.483 |
| | gemini-2-5-flash | 47 | 0.00% | 10.64% | 23.40% | 29.79% | 36.17% | 46.81% | 55.32% | 0.351 |
| | gemini-2-5-flash (search) | 47 | 2.13% | 14.89% | 17.02% | 25.53% | 36.17% | 42.55% | 48.94% | 0.272 |
| | o3 (high) | 47 | 2.13% | 17.02% | 29.79% | 34.04% | 36.17% | 40.43% | 48.94% | 0.425 |
| | o3 (high, search) | 47 | 2.13% | 21.28% | 31.91% | 34.04% | 38.30% | 42.55% | 51.06% | 0.414 |
| | o4-mini (high) | 47 | 2.13% | 10.64% | 17.02% | 21.28% | 23.40% | 31.91% | 44.68% | 0.401 |
| | o4-mini (high, search) | 43 | 2.33% | 13.95% | 18.60% | 25.58% | 30.23% | 37.21% | 44.19% | 0.340 |
| | gpt5 (high) | 47 | 4.26% | 19.15% | 23.40% | 34.04% | 38.30% | 40.43% | 48.94% | 0.249 |
| BILI | gpt-5 (high, search) | 46 | 2.17% | 21.74% | 28.26% | 30.43% | 36.96% | 41.30% | 58.70% | 0.275 |
| | gpt4-o | 46 | 0.00% | 10.87% | 21.74% | 26.09% | 28.26% | 36.96% | 52.17% | 0.273 |
| | gpt4-o (search) | 47 | 0.00% | 8.51% | 21.28% | 29.79% | 40.43% | 46.81% | 55.32% | 0.204 |
| | claude4-sonnet | 45 | 2.22% | 6.67% | 15.56% | 22.22% | 31.11% | 35.56% | 44.44% | 0.149 |
| | claude4-opus | 46 | 2.17% | 8.70% | 15.22% | 21.74% | 32.61% | 41.30% | 47.83% | 0.232 |
| | skywork-r1v3 | 47 | 0.00% | 2.13% | 6.38% | 17.02% | 29.79% | 38.30% | 53.19% | 0.192 |
| | GLM-4.5V | 47 | 2.13% | 8.51% | 17.02% | 23.40% | 29.79% | 38.30% | 51.06% | 0.268 |
| | gemini-2-5-pro | 93 | 58.06% | 73.12% | 77.42% | 77.42% | 80.65% | 83.87% | 86.02% | 0.814 |
| | gemini-2-5-pro (search) | 96 | 65.63% | 73.96% | 77.08% | 80.21% | 81.25% | 83.33% | 85.42% | 0.803 |
| | gemini-2-5-flash | 96 | 46.88% | 63.54% | 67.71% | 72.92% | 77.08% | 81.25% | 86.46% | 0.684 |
| | gemini-2-5-flash (search) | 96 | 57.29% | 68.75% | 70.83% | 70.83% | 73.96% | 76.04% | 81.25% | 0.665 |
| | o3 (high) | 95 | 54.74% | 70.53% | 72.63% | 73.68% | 76.84% | 76.84% | 84.21% | 0.686 |
| | o3 (high, search) | 96 | 55.21% | 66.67% | 68.75% | 71.88% | 71.88% | 71.88% | 73.96% | 0.789 |
| | o4-mini (high) | 96 | 27.08% | 44.79% | 48.96% | 55.21% | 61.46% | 63.54% | 68.75% | 0.652 |
| | o4-mini (high, search) | 93 | 52.69% | 56.99% | 60.22% | 63.44% | 65.59% | 67.74% | 70.97% | 0.572 |
| | gpt5 (high) | 95 | 50.53% | 68.42% | 72.63% | 72.63% | 76.84% | 76.84% | 81.05% | 0.521 |
| YT | gpt-5 (high, search) | 96 | 63.54% | 72.92% | 76.04% | 76.04% | 76.04% | 76.04% | 81.25% | 0.354 |
| | gpt4-o | 95 | 46.32% | 64.21% | 68.42% | 72.63% | 75.79% | 75.79% | 82.11% | 0.630 |
| | gpt4-o (search) | 95 | 47.37% | 63.16% | 68.42% | 70.53% | 75.79% | 76.84% | 81.05% | 0.492 |
| | claude4-sonnet | 92 | 29.35% | 43.48% | 46.74% | 52.17% | 54.35% | 57.61% | 68.48% | 0.491 |
| | claude4-opus | 89 | 39.33% | 49.44% | 51.69% | 56.18% | 61.80% | 64.04% | 70.79% | 0.540 |
| | skywork-r1v3 | 96 | 7.29% | 15.63% | 16.67% | 21.88% | 28.13% | 33.33% | 43.75% | 0.495 |
| | GLM-4.5V | 95 | 18.95% | 36.84% | 42.11% | 53.68% | 61.05% | 67.37% | 70.53% | 0.609 |

Table 10: Answer and thinking scores for VLMs on Bilibili and YouTube image source, with and without web search.

| VLMs | Gemini-2.5-pro | | Gemini-2.5-flash | | o3 (high) | | o4-mini (high) | | GPT5 (high) | | GPT4-o | | Claude4-Sonnet | Claude4-Opus | Skywork-R1V3 | GLM-4.5V |
|---|---|---|---|---|---|---|---|---|---|---|---|---|---|---|---|---|
| | No Web | Web | No Web | Web | No Web | Web | No Web | Web | No Web | Web | No Web | Web | No Web | No Web | No Web | No Web |
| | | | | | | | | | *Bilibili* | | | | | | | |
| Total Samples | 141 | 141 | 141 | 141 | 141 | 141 | 141 | 135 | 141 | 141 | 138 | 141 | 141 | 141 | 136 | 140 |
| Answer Score | 0.261 | 0.268 | 0.153 | 0.201 | 0.239 | 0.220 | 0.165 | 0.208 | 0.236 | **0.281** | 0.232 | 0.192 | 0.127 | 0.106 | 0.134 | 0.196 |
| Thinking Score | 0.520 | 0.459 | 0.418 | 0.370 | 0.481 | 0.548 | 0.382 | 0.347 | 0.375 | 0.310 | 0.325 | 0.232 | 0.210 | 0.223 | 0.197 | 0.317 |
| | | | | | | | | | *YouTube* | | | | | | | |
| Total Samples | 26 | 26 | 26 | 26 | 26 | 26 | 26 | 26 | 25 | 26 | 26 | 26 | 26 | 26 | 25 | 27 |
| Answer Score | 0.796 | 0.847 | 0.616 | 0.724 | 0.797 | **0.901** | 0.612 | 0.674 | 0.789 | 0.756 | 0.719 | 0.710 | 0.383 | 0.508 | 0.332 | 0.568 |
| Thinking Score | 0.762 | 0.742 | 0.636 | 0.644 | 0.646 | 0.675 | 0.644 | 0.606 | 0.499 | 0.315 | 0.685 | 0.509 | 0.468 | 0.522 | 0.511 | 0.663 |

## E.2 WHERESTREET

We first present two success cases using o4-mini with web (Table 14) and Gemini-2.5-pro without web (Table 15). Then, we present two failure cases for GLM-4.5-V (Table 16) and Gemini-2.5-pro with web (Table 17).

## F BENCHMARK SAMPLES

Here, we show three samples from GeoChain (Table 18,19, and 20). For GeoReasoner, the test set is not released[6]. Then, we show three samples from WhereStreet in Table 21, 22, and 23.

## G DECLARATION OF AI TOOL USAGE

During the preparation of this manuscript, we used OpenAI's GPT-5 model for minor language refinement and smoothing of the writing. The AI tool was not used for generating original content, conducting data analysis, or formulating core scientific ideas. All conceptual development, experimentation, and interpretation were conducted independently without reliance on AI tools.

---

[6]https://github.com/lingli1996/GeoReasoner/issues/3

Table 11: Ablation on reasoning effort and web search.

| | o3 | | | | | | o4-mini | | | | | | GPT5 | | | | | |
|---|---|---|---|---|---|---|---|---|---|---|---|---|---|---|---|---|---|---|
| | Low | | Medium | | High | | Low | | Medium | | High | | Low | | Medium | | High | |
| | No Web | Web | No Web | Web | No Web | Web | No Web | Web | No Web | Web | No Web | Web | No Web | Web | No Web | Web | No Web | Web |
| *Bilibili* | | | | | | | | | | | | | | | | | | |
| Total Samples | 139 | 141 | 141 | 141 | 141 | 141 | 140 | 140 | 141 | 141 | 141 | 135 | 141 | 138 | 141 | 141 | 141 | 141 |
| Answer Score | 0.257 | 0.235 | 0.262 | **0.268** | 0.239 | 0.220 | 0.197 | 0.152 | 0.175 | 0.198 | 0.165 | **0.208** | 0.261 | 0.254 | 0.233 | 0.265 | 0.236 | **0.281** |
| Thinking Score | 0.496 | 0.461 | 0.455 | 0.496 | 0.481 | 0.548 | 0.414 | 0.381 | 0.390 | 0.376 | 0.382 | 0.347 | 0.304 | 0.092 | 0.319 | 0.232 | 0.375 | 0.310 |
| *YouTube* | | | | | | | | | | | | | | | | | | |
| Total Samples | 26 | 26 | 26 | 26 | 26 | 26 | 25 | 26 | 26 | 26 | 26 | 26 | 26 | 26 | 26 | 26 | 25 | 26 |
| Answer Score | 0.843 | 0.772 | 0.739 | 0.797 | 0.797 | **0.901** | 0.627 | 0.636 | 0.654 | **0.729** | 0.612 | 0.674 | 0.769 | 0.819 | **0.831** | 0.699 | 0.789 | 0.756 |
| Thinking Score | 0.763 | 0.704 | 0.688 | 0.585 | 0.646 | 0.675 | 0.736 | 0.737 | 0.661 | 0.625 | 0.644 | 0.606 | 0.334 | 0.179 | 0.288 | 0.223 | 0.499 | 0.315 |

Table 12: Error analysis breakdown (%). We report the frequency of missed visual clues, overthinking, and search-related errors across different models.

| Model | Missed Visual Clues | Overthinking | Incomplete/ Search |
|---|---|---|---|
| o4-mini (search, high) | 52.19 | 1.35 | 49.49 |
| o3 (high) | 43.04 | 2.91 | 38.51 |
| Gemini-2.5-pro | 37.10 | 0.97 | 54.84 |
| Gemini-2.5-pro (search) | 35.16 | 2.90 | 54.52 |
| GLM-4.5V | 63.43 | 34.95 | 7.44 |
| Skywork-R1V3 | 76.97 | 35.86 | 48.68 |

Table 13: Case Study for `WhereCountry`: GPT-4o without web access scrutinizes tree types and fencing style to narrow to Chile.

**User**

**Question:** Look at this image and determine which country it was taken in. Think first, then choose from the following options: [Argentina, Chile, Bolivia, Peru].
**Ground Truth Answer:** Chile

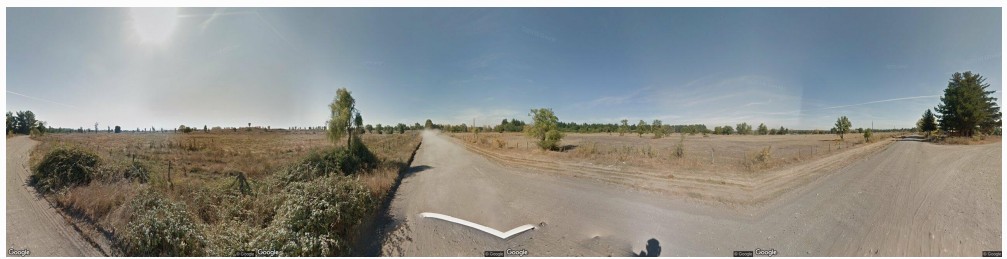

**GPT-4o (no web)**

This image showcases a countryside landscape with flat terrain, scattered vegetation, and dusty roads. The environment is characteristic of regions in South America, but notably it resembles parts of southern Chile. The tree types and fencing style suggest a rural area typical of Chilean farmland rather than the more elevated terrains of Bolivia, Peru, or northern Argentina. **Final answer: Chile**

**GPT-4o (with web)**

The landscape appears to be flat and dry, typical of areas found in Argentina and Chile. The road is unpaved, and there are open fields and sparse trees, which can be found in the Pampas region of Argentina or certain areas of Chile. However, without distinctive architectural or signage clues, determining the exact country can be challenging. The prevalence of this type of landscape in Argentina suggests it might be from there. **Final answer: Argentina**

Table 14: Case Study for `WhereStreet` with o4-mini (web).

**System:** You are a geolocation analyst. Given an image, a **HINT**, and an **ANSWER_TYPE**, do the following:

1. Extract concrete visual evidence (e.g., signage text/language, road markings, license-plate style, driving side, architecture, vegetation/biome, terrain, rail features, utility furniture).

2. Reason via a coarse → fine funnel (country → region → city → street) and commit to *one* location at the requested granularity.

3. Use Google Search tool if needed.

If a finer granularity is requested, you *must* choose a plausible candidate at that level rather than stopping early. If uncertain, still pick the single best candidate matching the ANSWER-TYPE. Respond in English. Provide detailed reasoning between the `<think> </think>` tags and the final answer between the `<answer> </answer>` tags.

**User**

**Question:** Where was this image taken based on visual clues and the provided hint?
**Hint:** None.
**Answer type:** street
**Ground-truth answer:** Chimney Rock at Chimney Rock State Park, Asheville, North Carolina, USA

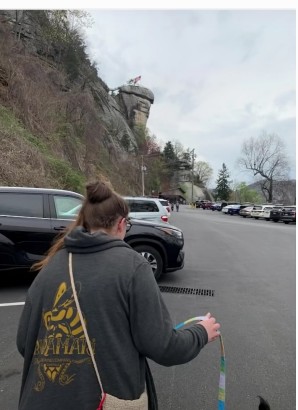

**o4-mini (web) Response** We see the distinctive 315-foot granite monolith topped by an American flag and the adjacent visitor parking lot. This is Chimney Rock at Chimney Rock State Park, accessible via Main Street in Chimney Rock Village just off NC-9. The attraction's own information gives a primary GPS approach point as *665 Main Street, Chimney Rock, NC*, with coordinates approximately $35.432799, -82.250260$. The official state parks site also lists the park office location at *743 Chimney Rock Park Road* ($35.4327, -82.2502$), but the temporary bridge entrance and main lot seen in the photo correspond to the Main Street address and the more precise coordinates.

**Final answer: 665 Main Street, Chimney Rock, NC 28720, USA** ($35.432799, -82.250260$)

Table 15: Case Study for `WhereStreet` with Gemini-2.5-pro (no web).

**System:** You are a geolocation analyst. Given an image, a **HINT**, and an **ANSWER_TYPE**, do the following:

1. Extract concrete visual evidence (e.g., signage text/language, road markings, license-plate style, driving side, architecture, vegetation/biome, terrain, rail features, utility furniture).

2. Reason via a coarse → fine funnel (country → region → city → street) and commit to *one* location at the requested granularity.

3. Use Google Search tool if needed.

If a finer granularity is requested, you *must* choose a plausible candidate at that level rather than stopping early. If uncertain, still pick the single best candidate matching the ANSWER-TYPE. Respond in English. Provide detailed reasoning between the `<think> </think>` tags and the final answer between the `<answer> </answer>` tags.

**User**

**Question:** Where was this image taken based on visual clues and the provided hint?
**Hint:** The image is taken in China.
**Answer type:** county
**Ground-truth answer:** Meilan District, Haikou City, Hainan, China

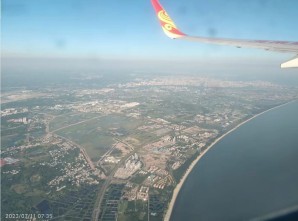

Table 15: Case Study for `WhereStreet` with Gemini-2.5-pro (no web) (continued).

---

**Gemini-2.5-pro (no web) Response**

1. **Initial Analysis of the Image & Hint:**

   - The image is an aerial photograph taken from an airplane window.
   - The HINT states the location is in China.
   - A timestamp in the bottom-left corner reads "2023/07/11 07:35".
   - A prominent feature is the airplane's winglet, which is red with a golden/yellow swirling pattern. This is the distinctive livery of Hainan Airlines (海南航空).
   - The landscape below is a coastal region with a long, straight, sandy coastline on the right side of the image.
   - Inland, there is a mix of urban development, green areas, and numerous rectangular, water-filled plots characteristic of aquaculture farms (fish or shrimp ponds).
   - In the middle-ground to background, a large airport with at least two parallel runways is visible.
   - Beyond the airport, a large, dense urban area (a major city) stretches across the horizon.
   - Given the timestamp (07:35 in July), the lighting suggests the sun is in the east. The scene appears to be illuminated from the right and slightly behind the plane, indicating the aircraft is likely flying in a generally northerly or westerly direction.

2. **Coarse-to-Fine Geolocation Funnel:**

   - **Country:** China (confirmed directly by the hint).
   - **Province/Region:** The combination of:
     - Hainan Airlines as the carrier,
     - a tropical/subtropical coastal environment,
     - extensive aquaculture ponds near the coast, and a major city with a large airport

   strongly points toward Hainan Province. Hainan Airlines' primary hub is Haikou Meilan International Airport (HAK) in Haikou, the provincial capital. Sanya, on the south coast, is another major city with an airport (SYX).

   - **City Candidates (Haikou vs. Sanya):**
     - Haikou is on the north coast of Hainan Island. Haikou Meilan International Airport (HAK) is located southeast of the main urban core and about 15 km inland from the northern coast.
     - Sanya is on the south coast. Sanya Phoenix International Airport (SYX) is located directly adjacent to the coastline, west of the main city.
     - The image shows the airport significantly inland from the coast, with a large city even farther inland. This layout is much more consistent with Haikou than with Sanya. In Sanya, the airport sits right on the coast.
   - **Flight-Path Hypothesis (Haikou):**
     - Consider an approach to Haikou Meilan International Airport (HAK) from the east or southeast for a westerly/northwesterly landing.
     - In such a scenario, the coastline of the Qiongzhou Strait would lie to the right (north), the airport would be ahead and slightly to the left (southwest), and the main urban area of Haikou would appear farther inland to the left (west).
     - This arrangement matches the spatial configuration in the image: long coastline on the right, aquaculture belt inland, airport in the mid-ground, and dense city on the horizon.

---

Table 15: Case Study for `WhereStreet` with Gemini-2.5-pro (web) (continued).

---

**Gemini-2.5-pro (web) Response**

3. **Pinpointing the Location (County / District Level):**

- The requested `ANSWER_TYPE` is `county`. In the Chinese administrative system, for major cities, an urban "District" (区, qū) is effectively a county-level unit (equivalent to a "County" (县, xiàn).
- The area under the aircraft — including Haikou Meilan International Airport and the coastal plain to its northeast — belongs to Meilan District (美兰区) of Haikou City.
- Satellite imagery of Haikou Meilan International Airport (HAK) confirms that it has two parallel runways, matching the airport visible in the photograph.
- The coastline east/northeast of Haikou within Meilan District forms a long, sandy shore, consistent with the straight beach line on the right side of the image.
- Between the coast and the airport, especially around towns such as Yanfeng (演丰镇), there are extensive aquaculture ponds. This pattern of rectangular, water-filled plots coincides with what is seen in the aerial photograph.
- The dense urban area in the background corresponds to the main city of Haikou, while newer high-rise developments closer to the coast align with development zones such as Guilinyang (桂林洋).

4. **Conclusion:**

- All available visual evidence — airline livery (Hainan Airlines), coastal geography, aquaculture layout, airport configuration, and urban morphology — consistently points to the coastal area of Meilan District in Haikou, Hainan Province.
- The airplane is likely on approach to Haikou Meilan International Airport (HAK), flying roughly northwest along the coast of the Qiongzhou Strait.
- Therefore, at the requested county-level granularity, the best and most consistent location is Meilan District.

**Final answer: Meilan District (美兰区), Haikou, Hainan Province, China**

---

Table 16: Case Study for `WhereStreet` with GLM-4.5-V (no web). In the response, the model has proposed the assumption of the correct final answer, but self-reject it.

---

**System:** You are a geolocation analyst. Given an image, a **HINT**, and an **ANSWER_TYPE**, do the following:

1. Extract concrete visual evidence (e.g., signage text/language, road markings, license-plate style, driving side, architecture, vegetation/biome, terrain, rail features, utility furniture).

2. Reason via a coarse → fine funnel (country → region → city → street) and commit to *one* location at the requested granularity.

If a finer granularity is requested, you *must* choose a plausible candidate at that level rather than stopping early. If uncertain, still pick the single best candidate matching the ANSWER-TYPE. Respond in English. Provide detailed reasoning between the `<think> </think>` tags and the final answer between the `<answer> </answer>` tags.

**User**

**Question:** Where was this image taken based on visual clues and the provided hint?
**Hint:** This image is likely taken in China.
**Answer type:** City
**Ground-truth answer:** Dalian City, Liaoning, China.

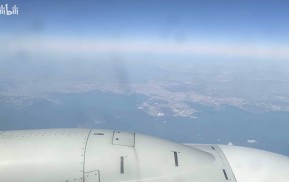

---

**GLM-4.5-V (no web) Response**

`<think>` Got it, let's try to figure out the city here. First, the image is taken from an airplane, so we're looking at an aerial view. The hint says it's likely in China. Let's think about Chinese cities with distinctive water bodies or coastlines. The image shows a large body of water with a city on the other side. Maybe a bay or a strait? Wait, the timestamp is 2022.12.06 11:45. Let's think about Chinese cities that have such a landscape. Maybe Xiamen? Because Xiamen has a bay with islands, like Gulangyu Island. Or maybe Qingdao? But Xiamen's geography is more like a bay with a city on the mainland and islands. Wait, the image shows a large water body with a city spread out. **Another possibility is Dalian, but Dalian's coastline is different.** Wait, maybe it's Xiamen. Let me check: Xiamen is in Fujian province, and it has Xiamen Bay, with the city on the mainland and islands like Gulangyu. The aerial view might show that. Since the hint says China, and we need a city, Xiamen is a plausible candidate. I think that's the best guess here.`</think>` `<|begin_of_box|><thinking>`
1. Visual Evidence Analysis: The image shows an aerial view from an airplane, with a large body of water (likely a bay or strait) and a densely populated urban area on the opposite shore. The timestamp is 2022.12.06 11:45, and the hint indicates the location is likely in China.
2. Coarse→Fine Funnel: - Country: The hint explicitly states likely taken in China,šo we start with China. - Region/Province: Chinese coastal cities with prominent bays or straits are candidates. Fujian province has several such cities, including Xiamen, which is known for its bay (Xiamen Bay) and islands (e.g., Gulangyu Island). - City: Xiamen's geography matches the image: a large water body with a city spread across the mainland and nearby islands. The aerial view of Xiamen Bay, with the urban area and water features, aligns with the visual evidence.
3. Conclusion: Based on the coastal geography, urban layout, and hint, Xiamen is the most plausible city. `</thinking>` **Final answer: Xiamen**

Table 17: Case Study for `WhereStreet` with Gemini-2.5-pro (web).

**System:** You are a geolocation analyst. Given an image, a **HINT**, and an **ANSWER_TYPE**, do the following:

1. Extract concrete visual evidence (e.g., signage text/language, road markings, license-plate style, driving side, architecture, vegetation/biome, terrain, rail features, utility furniture).

2. Reason via a coarse $\rightarrow$ fine funnel (country $\rightarrow$ region $\rightarrow$ city $\rightarrow$ street) and commit to *one* location at the requested granularity.

3. Use Google Search tool if needed.

If a finer granularity is requested, you *must* choose a plausible candidate at that level rather than stopping early. If uncertain, still pick the single best candidate matching the ANSWER-TYPE. Respond in English. Provide detailed reasoning between the `<think> </think>` tags and the final answer between the `<answer> </answer>` tags.

**User**

**Question:** Where was this image taken based on visual clues and the provided hint?
**Hint:** The photograph was taken in the mid-1950s in Northern Europe or south of the UK.
**Answer type:** Latitude & Longitude
**Ground-truth answer:** 50.7999, -0.5425

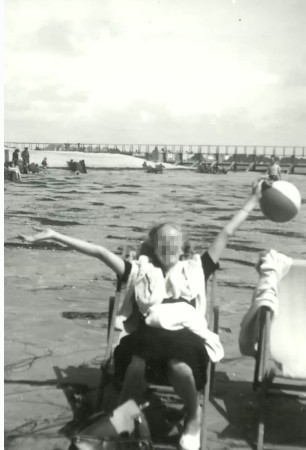

Table 17: Case Study for `WhereStreet` with Gemini-2.5-pro (web) (continued).

**Gemini-2.5-pro (web) Response**

1. **Initial Analysis of Visual Evidence:**
   - The image provided is a black-and-white photograph depicting a person sitting on a deckchair on a wide sandy beach.
   - The background shows many other people on the beach, also using deckchairs.
   - A prominent feature in the background is a long, continuous seawall or promenade with a simple railing. Behind the promenade, some buildings are faintly visible.
   - The overall atmosphere, clothing, and the presence of deckchairs suggest a popular seaside resort.
   - The sand appears dark and possibly wet, indicating it might be low tide on a tidal flat beach.

2. **Deconstruction of the HINT:**
   - The HINT states the photo was taken in the mid-1950s. This aligns with the style of the photograph and the clothing.
   - The location is given as Northern Europe or south of the UK. This narrows the search area to the coastlines of the North Sea and the English Channel. Countries to consider include the UK (south coast), France (north coast), Belgium, the Netherlands, and possibly Germany or Denmark.

3. **Coarse-to-Fine Geolocation Strategy:**
   - **Country Level:** The combination of a very wide, flat, sandy beach and a long, man-made seawall/promenade (dijk or esplanade) is highly characteristic of the Belgian and Dutch coasts. While southern UK and northern France have promenades, the continuous, uniform nature seen here is particularly emblematic of the Belgian coastline, which is almost entirely lined with such a structure to protect the low-lying land.
   - **City/Region Level:** I will focus on major seaside resorts along the Belgian and Dutch coasts from the 1950s. Key candidates include Ostend (Oostende), Blankenberge, and Knokke in Belgium, and Scheveningen or Zandvoort in the Netherlands. The structure in the photo appears to be a concrete wall with a simple metal railing. I will use search queries to find historical photos of these locations to compare the promenades.
   - **Street/Specific Location Level:** By comparing the promenade's architecture and any visible background buildings, I will attempt to pinpoint a specific city and then a segment of the beach.

4. **Execution of Search and Verification:**
   - I will perform Google searches using terms like "Ostend beach 1950s", "Blankenberge Zeedijk 1950s", "Scheveningen pier 1950s", and "Belgian coast promenade vintage photo".

   - Numerous historical photos of the beach at Ostend, Belgium, show a scene identical in character to the provided image: a vast expanse of sand, crowds with deckchairs, and the long Albert I-Promenade in the background. The style of the simple railing seen in the image is consistent with historical photos of the Ostend seawall from that period.
   - Reverse image searches and further specific searches for "Oostende strand 1955" confirm that this photograph is widely identified as being taken on the beach at Ostend, Belgium.
   - To find the coordinates, I need to identify a plausible spot on this long beach. The beach at Ostend is called the "Groot Strand." It stretches along the Albert I-Promenade. A very central and popular spot for beachgoers would be in front of the main landmark, the Casino Kursaal. The faint outlines of buildings in the background are consistent with the general appearance of the hotel and apartment facades along this promenade.
   - Therefore, I will select a point on the sand of the Groot Strand, facing the Casino Kursaal, as the most probable location for this photograph. The coordinates of the Casino Kursaal are approximately 51.2320° N, 2.9115° E. The photo would have been taken from the beach, looking towards the casino.

Table 17: Case Study for `WhereStreet` with Gemini-2.5-pro (web) (continued).

---

**Gemini-2.5-pro (web) Response**

1. **Final Coordinate Selection:**

   • A point on the beach, a reasonable distance from the promenade, looking towards the center of Ostend's seafront, provides the best estimate. I will select coordinates on the sand directly offshore from the Casino Kursaal. This leads to approximately 51.2325° N, 2.9108° E.

**Final answer: 51.2325° N, 2.9108° E**

---

Table 18: Example QA instance from GeoChain (Mapillary key: `E-fIB3K6E-Z-GSW_xI-rSw`).

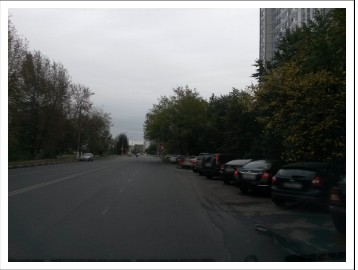

| Field | Value |
|---|---|
| Key | `E-fIB3K6E-Z-GSW_xI-rSw` |
| City | Moscow |
| Country | Russian Federation |
| Latitude, Longitude | 55.6464, 37.7250 |
| Locatability score | 0.462 |

| Rank | Diff. | Question | Answer |
|---|---|---|---|
| 1 | Easy | Do you see any boats or ships? | No |
| 2 | Easy | Do you see one or more of the following vehicles: Bus, Truck, Car, Van, Motorbike, Minibike, Bicycle? | Yes |
| 3 | Easy | Can you see any traffic lights? | Yes |
| 4 | Easy | Can you see any flag? | No |
| 5 | Easy | Would you say this location is near the Equator? | No |
| 6 | Easy | Does this location seem to be close to the Poles? | No |
| 7 | Easy | Is this place located in the Northern Hemisphere? | Yes |
| 8 | Easy | Which continent best describes where this location is? | Europe |
| 9 | Medium | What side of the road do vehicles drive on here? | Right |
| 10 | Medium | What country is this place located in? | Russian Federation |
| 11 | Medium | Is this place near coast? | No |
| 12 | Medium | Does this location appear to be an island? | No |
| 13 | Easy | Is this place located in a desert region? | No |
| 14 | Easy | Does this location seem to be in a mountainous or hilly region? | No |
| 15 | Medium | What is the most likely climate type for this location? | Continental |
| 16 | Easy | Does this place look like a big city? | Yes |
| 17 | Medium | Would you classify this place as a small town? | No |
| 18 | Hard | What language(s) are most likely spoken at this place? | Russian |
| 19 | Hard | Can you name the state or province this place belongs to? | Moscow |
| 20 | Hard | What is the name of the city, town, or village seen here? | Moscow |
| 21 | Hard | Based on everything observed, what are the latitude and longitude coordinates of this place? Please give a tuple of float coordinates (lat, lon). | 55.6464, 37.7250 |

Table 19: Example QA instance from GeoChain (Mapillary key: `9lTncS0AZRczU-PCGI4vQg`).

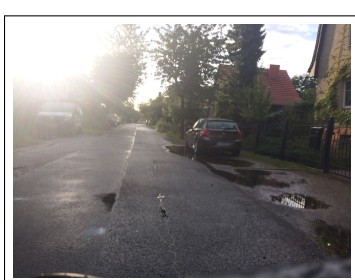

| Field | Value |
| --- | --- |
| Key | `9lTncS0AZRczU-PCGI4vQg` |
| City | Berlin |
| Country | Germany |
| Latitude, Longitude | 52.6003, 13.4591 |
| Locatability score | 0.598 |

| Rank | Diff. | Question | Answer |
| --- | --- | --- | --- |
| 1 | Easy | Do you see any boats or ships? | No |
| 2 | Easy | Do you see one or more of the following vehicles: Bus, Truck, Car, Van, Motorbike, Minibike, Bicycle? | Yes |
| 3 | Easy | Can you see any traffic lights? | No |
| 4 | Easy | Can you see any flag? | No |
| 5 | Easy | Would you say this location is near the Equator? | No |
| 6 | Easy | Does this location seem to be close to the Poles? | No |
| 7 | Easy | Is this place located in the Northern Hemisphere? | Yes |
| 8 | Easy | Which continent best describes where this location is? | Europe |
| 9 | Medium | What side of the road do vehicles drive on here? | Right |
| 10 | Medium | What country is this place located in? | Germany |
| 11 | Medium | Is this place near coast? | No |
| 12 | Medium | Does this location appear to be an island? | No |
| 13 | Easy | Is this place located in a desert region? | No |
| 14 | Easy | Does this location seem to be in a mountainous or hilly region? | No |
| 15 | Medium | What is the most likely climate type for this location? | Temperate |
| 16 | Easy | Does this place look like a big city? | Yes |
| 17 | Medium | Would you classify this place as a small town? | No |
| 18 | Hard | What language(s) are most likely spoken at this place? | German |
| 19 | Hard | Can you name the state or province this place belongs to? | Berlin |
| 20 | Hard | What is the name of the city, town, or village seen here? | Berlin |
| 21 | Hard | Based on everything observed, what are the latitude and longitude coordinates of this place? Please give a tuple of float coordinates (lat, lon). | 52.6003, 13.4591 |

Table 20: Example QA instance from GeoChain (Mapillary key: `w1SOt-lYqc5cfhDWop2Beg`).

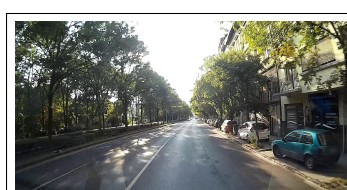

| Field | Value |
|---|---|
| Key | `w1SOt-lYqc5cfhDWop2Beg` |
| City | Budapest |
| Country | Hungary |
| Latitude, Longitude | 47.5091, 19.0166 |
| Locatability score | 0.472 |

| Rank | Diff. | Question | Answer |
|---|---|---|---|
| 1 | Easy | Do you see any boats or ships? | No |
| 2 | Easy | Do you see one or more of the following vehicles: Bus, Truck, Car, Van, Motorbike, Minibike, Bicycle? | Yes |
| 3 | Easy | Can you see any traffic lights? | No |
| 4 | Easy | Can you see any flag? | No |
| 5 | Easy | Would you say this location is near the Equator? | No |
| 6 | Easy | Does this location seem to be close to the Poles? | No |
| 7 | Easy | Is this place located in the Northern Hemisphere? | Yes |
| 8 | Easy | Which continent best describes where this location is? | Europe |
| 9 | Medium | What side of the road do vehicles drive on here? | Right |
| 10 | Medium | What country is this place located in? | Hungary |
| 11 | Medium | Is this place near coast? | No |
| 12 | Medium | Does this location appear to be an island? | No |
| 13 | Easy | Is this place located in a desert region? | No |
| 14 | Easy | Does this location seem to be in a mountainous or hilly region? | No |
| 15 | Medium | What is the most likely climate type for this location? | Continental |
| 16 | Easy | Does this place look like a big city? | Yes |
| 17 | Medium | Would you classify this place as a small town? | No |
| 18 | Hard | What language(s) are most likely spoken at this place? | Hungarian |
| 19 | Hard | Can you name the state or province this place belongs to? | Budapest |
| 20 | Hard | What is the name of the city, town, or village seen here? | Budapest |
| 21 | Hard | Based on everything observed, what are the latitude and longitude coordinates of this place? Please give a tuple of float coordinates (lat, lon). | 47.5091, 19.0166 |

Table 21: Example from `WhereStreet`.

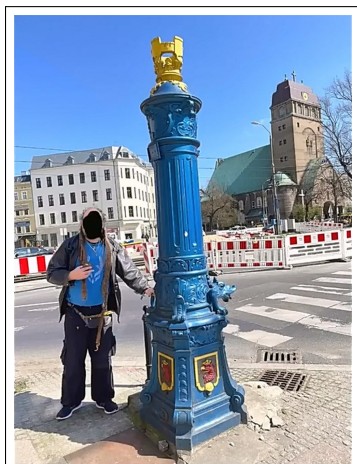

| Field | Value |
|---|---|
| Answer type | latitude & longitude |
| Coordinates | 53.4258, 14.5457 |
| Key clues | • A coat of arms on the pump features a red griffin's head with a crown on a blue background.
• A large church with a distinct blocky tower and a green roof is visible in the background.
• Tram tracks run along the street. |
| Hint | The street shop on the left of the image might have the sign saying *"KANCELARIA"* and *"RACHUNKOWE"*. |

Table 22: Example from `WhereStreet`.

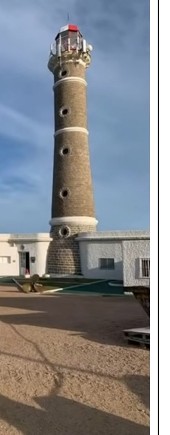

| Field | Value |
|---|---|
| Answer type | latitude & longitude |
| Coordinates | -34.8460, -54.6329 |
| Key clues | • Lighthouse and Uruguay flag in the photograph.
• The lighthouse's features, including the round windows. |
| Hint | No additional hint is provided for this sample. |

Table 23: Example from `WhereStreet`.

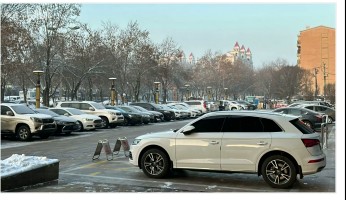

| Field | Value |
|---|---|
| Answer type | city |
| City (ground truth) | Changji Hui Autonomous Prefecture, Xinjiang Uyghur Autonomous Region, China |
| Key clues | • The presence of snow and bare deciduous trees.
• A distinctive European-style building with multiple red, pointed roof.
• A sign on a lamppost with the number 1054 is identified as a "lamppost police reporting number". |
| Hint | This image is taken in China. |

