# OpenReview forum: "Where in the World? A Vision-Language Benchmark for Probing Model Geolocation Skills Across Scales"
_ICLR.cc/2026/Conference — Submitted to ICLR 2026_

### Official Review · Reviewer_iQxJ · 2025-10-27

**Soundness:** 2
**Presentation:** 2
**Contribution:** 1
**Rating:** 2
**Confidence:** 5

**Summary:**

This paper introduces WhereBench, a new benchmark to evaluate the geolocation capabilities of Vision-Language Models (VLMs). The benchmark is split into two tasks: WhereCountry, a coarse-grained multiple-choice task, and WhereStreet, a fine-grained, reasoning-based task. The paper's primary contribution is a novel "Shapley-reweighted thinking score," designed to evaluate the faithfulness of a model's reasoning process by checking its use of human-verified visual clues. The authors benchmark 12 VLMs, finding that closed-source models dominate, web search provides mixed results, and models exhibit strong regional biases.

**Strengths:**

1. The primary strength is the introduction of a novel and potentially useful "thinking score" metric, reweighted by Shapley values, to evaluate the process of reasoning rather than just the final answer's accuracy.
2. The paper's experiments provide a nuanced analysis of when web search and deeper reasoning can help or, counter-intuitively, hurt localization performance depending on the task's nature.

**Weaknesses:**

1. The central claim that "there remains a lack of a fair and comprehensive benchmark that evaluates... faithfulness"  is incorrect. Several such benchmarks have been proposed at a much larger scale, including GeoChain (Yerramilli et al., 2025) and Gaea (Campos et al., 2025), which the paper fails to cite or compare against.
2. The benchmark dataset is extremely small, with only 810 total examples (500 for WhereCountry, 310 for WhereStreet), making it difficult to draw statistically significant or generalizable conclusions.
3. The WhereStreet task has a severe, built-in geographic bias, as it is sourced only from English- and Chinese-language videos. This is a major design flaw that limits the benchmark's diversity and directly leads to the "regional bias"  reported as a finding.
4. The paper's annotation protocol is underspecified. It fails to mention if inter-annotator agreement was measured for the 861 key clues, or how ambiguities between the 7 annotators were resolved. This undermines the reliability of the ground-truth clues used for the novel "thinking score."
5. The paper misinterprets its own results regarding WhereCountry. The claim that the task is "less reasoning-intensive"  is likely incorrect; it is more probable that the models' poor visual perception is the bottleneck, preventing them from knowing what to search for or reason about, a common failure mode in multimodal tasks.
6. The methodology for the novel Shapley-reweighted score is underspecified. The paper states $v(S)$ is estimated by "prompting the judge (Gemini-2.5-Pro) for all $2^{|C|}$ subsets" but never states the average number of clues $|C|$, which could make this computationally infeasible. The value function $v(S)$ ("achievable answer quality") is also highly subjective and not clearly defined.
7. The paper validates its answer-scoring judge but provides no such validation for its clue-in-reasoning judge. It is unclear how the judge reliably distinguishes a clue that was truly "used" from one that was merely "mentioned."The paper's regional bias finding may be based on confounding variables. It is unclear if the performance gap is due to model training data bias or other factors, such as a different density/type of visual clues (e.g., Roman vs. Chinese script) in the two biased data sources.

**Questions:**

1. What was the inter-annotator agreement (IAA) for the 861 verified key clues, and what protocol was used by the 7 annotators to resolve disagreements?
2. What is the average number of clues $|C|$ per sample, and how exactly is the "achievable answer quality" score for $v(S)$ defined and prompted to the judge?

---

> ### Author Response · Authors · 2025-11-27
>
> ### W1.
> > The central claim that "there remains a lack of a fair and comprehensive benchmark that evaluates... faithfulness" is incorrect. Several such benchmarks have been proposed at a much larger scale, including GeoChain (Yerramilli et al., 2025) and Gaea (Campos et al., 2025), which the paper fails to cite or compare against.
>
> Thank you for identifying these additional references. We kindly refer the reviewer to General Response section 3 and Related Work (sec2) in the updated manuscript, where we provide a detailed feature comparison table between previous works and our WhereBench.
>
> ### W2.
> >The benchmark dataset is extremely small, with only 810 total examples (500 for WhereCountry, 310 for WhereStreet), making it difficult to draw statistically significant or generalizable conclusions.
>
> We appreciate your concern regarding the dataset scale. We kindly refer to General Response section 1, where we address the statistically significant concerns and we are expanding the dataset actively.
>
> ### W3.
> >The WhereStreet task has a severe, built-in geographic bias, as it is sourced only from English- and Chinese-language videos. This is a major design flaw that limits the benchmark's diversity and directly leads to the "regional bias" reported as a finding.
>
> We respectfully clarify a misunderstanding regarding the data source. The language of the source video (English/Chinese) dictates the narration, not the geographic location of the image. As shown in Figure 2a (Global Map) and Figure 2b (Distribution), our benchmark includes coverage of locations in South America, Africa, Europe, and Oceania, far beyond just English- or Chinese-speaking regions. Therefore, the source language does not inherently limit the geographic diversity of the visual content.
> We are also expanding our benchmark with additional data as detailed in General Response section 1. We will update the statistics once it is finished.
>
>
> ### W4.Q1.
> > The paper's annotation protocol is underspecified. It fails to mention if inter-annotator agreement was measured for the 861 key clues, or how ambiguities between the 7 annotators were resolved. This undermines the reliability of the ground-truth clues used for the novel "thinking score."
>
> > What was the inter-annotator agreement (IAA) for the 861 verified key clues, and what protocol was used by the 7 annotators to resolve disagreements?
>
> Thanks for this question regarding our annotation protocol. We have updated Appendix C.2 to provide a detailed description of the quality control measures. Specifically, we adopted a hierarchical verification procedure rather than a standard consensus voting mechanism such as IAA. Seven trained annotators processed distinct batches of data. Any ambiguous visual clues were escalated to a lead annotator for final decision.
> To ensure high fidelity, the lead expert performed random spot-checks (10 samples per annotator). If any error was found during spot-checks, that annotator's entire batch underwent a full re-verification. This rigorous verification process ensures the annotation's reliability.
>
> ### W5.
> >The paper misinterprets its own results regarding WhereCountry. The claim that the task is "less reasoning-intensive" is likely incorrect; it is more probable that the models' poor visual perception is the bottleneck, preventing them from knowing what to search for or reason about, a common failure mode in multimodal tasks.
>
> We appreciate the reviewer’s alternative interpretation regarding a "perception bottleneck." We agree that this is a possible explanation. However, we respectfully note that given the "black-box" nature of these proprietary models, definitively distinguishing between perception and reasoning is deductive for both interpretations. We cannot verify if the model failed to see the visual feature or if it saw the feature but failed to link it to the correct location in the WhereCountry subset.
>
> Rather than committing to a single unproven mechanism, we argue that the two factors are functionally interdependent. Our empirical data shows that reasoning compute plateaus on this task. Whether this is because the task inherently requires less reasoning, or because a perception bottleneck starves the reasoning module of valid inputs, the functional outcome is the same: **scaling test-time compute provides diminishing returns**.
> To reflect this necessary scientific caution, we have revised Section 4.1 in the manuscript to avoid making a definitive conclusion.

---

> > ### Author Response · Authors · 2025-11-27
> >
> > ### W6. W7. Q2.
> > >The methodology for the novel Shapley-reweighted score is underspecified. The paper states is estimated by "prompting the judge (Gemini-2.5-Pro) for all subsets" but never states the average number of clues , which could make this computationally infeasible. The value function  ("achievable answer quality") is also highly subjective and not clearly defined.
> >
> > >The paper validates its answer-scoring judge but provides no such validation for its clue-in-reasoning judge. It is unclear how the judge reliably distinguishes a clue that was truly "used" from one that was merely "mentioned."
> >
> > > What is the average number of clues  per sample, and how exactly is the "achievable answer quality" score for  defined and prompted to the judge?
> >
> > Thanks for your question. With 861 clues for 310 samples in WhereStreet, each sample has an average of 2.78 key clues. We kindly refer to General Response section 2 for the other concerns, where we include more details about the value function and our experiments on validating the clue-in-reasoning judge. We respectfully point to Table 6, where we rigorously defined the judge's protocol. The prompt explicitly instructs the model to classify a clue as "Used" only if there is a causal link to the final answer and to reject clues that are merely "Mentioned" or "Dismissed."
> >
> >
> > ### W7.
> > >The paper's regional bias finding may be based on confounding variables. It is unclear if the performance gap is due to model training data bias or other factors, such as a different density/type of visual clues (e.g., Roman vs. Chinese script) in the two biased data sources.
> >
> > We appreciate the reviewer’s hypothesis that "clue density" could be a confounding variable. On further reflection, we realized that our original regional bias claims risk overstating what our current data can support: disentangling the influence of training data, model origin, question difficulty, and benchmark composition would require much more controlled evidence than we have. In the interest of precision and avoiding speculative causal claims, we have removed this specific interpretation from the revision.

---

> > > ### Comment · Reviewer_iQxJ · 2025-11-28
> > >
> > > Thank you for your responses and clarifications. After considering your rebuttal, I have decided to maintain my original score for the submission.

---

### Official Review · Reviewer_A8fX · 2025-10-27

**Soundness:** 2
**Presentation:** 2
**Contribution:** 2
**Rating:** 2
**Confidence:** 5

**Summary:**

This paper introduces WhereBench, a novel benchmark designed to evaluate the image-grounded geolocation capabilities of Vision-Language Models (VLMs) across different geographic scales. The benchmark is comprised of two distinct tasks: WhereCountry, a multiple-choice question-answering task with 500 panoramic images for coarse, country-level localization, and WhereStreet, a more challenging task with 310 images requiring precise, street-level localization that often necessitates multi-step reasoning and the use of external tools like web search.

A key contribution of this work is the introduction of a novel evaluation metric, the "thinking score," which moves beyond simple accuracy to assess the quality of a model's reasoning process by measuring its ability to identify and utilize relevant visual cues.  The authors conduct a comprehensive evaluation of 12 state-of-the-art VLMs on WhereBench, analyzing performance across various settings (e.g., with and without tool use, varying reasoning lengths). The results reveal several key insights, including a significant performance gap between closed-source and open-source models, the surprising ineffectiveness of web search in certain scenarios, and strong regional biases in model performance.

**Strengths:**

1. **Important and Well-Motivated Problem:** The paper addresses a challenging and practically relevant problem. Image geolocation has numerous real-world applications, from aiding in search and rescue operations to enhancing content moderation systems.  The lack of a comprehensive benchmark for this task represents a significant gap in the literature, which this work effectively addresses.

2. **Thorough Experimental Suite:** The paper provides a rigorous and wide-ranging experimental evaluation, testing a diverse set of 12 VLMs across various settings (e.g., with and without tool use, varying reasoning lengths).

**Weaknesses:**

1. **Marginal Novelty and Missing Citations:** The paper's novelty is somewhat diminished by its close resemblance to recent works like GeoChain and GeoReasoner. These papers explore similar themes of geolocation reasoning with VLMs, yet they are not cited in the related works section. Acknowledging and differentiating this work from these existing papers is crucial for establishing its unique contribution.

2. **Limited Scale of the Dataset:** While well-designed, the WhereBench dataset is relatively small, with only 500 images for WhereCountry and 310 for WhereStreet.  This limited scale raises concerns about the statistical significance and generalizability of the findings. For a benchmark paper, a larger and more diverse dataset would be necessary to provide a more robust and reliable evaluation of model performance.

3. **Flawed Assumption about Web Search:** The finding that web search does not consistently improve accuracy is presented as a surprising result.  However, this is expected given that current web search APIs used by VLMs do not support visual search (i.e., searching with the image itself). The models are limited to searching with textual keywords derived from the image. If these keywords are incorrect or insufficient, the web search will only add noise to the context, potentially degrading performance.  This limitation of the underlying tool, rather than a deficiency in the models' reasoning, is a more likely explanation for the observed results.

4. **Dataset Bias and Lack of Transparency:** The WhereStreet dataset is heavily skewed towards Asia, which could introduce regional biases into the evaluation. Additionally, the paper is unclear about the source of the images used in the WhereCountry dataset.  Providing more transparency about the data sourcing and composition would improve the benchmark's credibility.

5. **Lack of Qualitative Analysis:** The paper would be significantly strengthened by a more in-depth qualitative analysis of the models' reasoning processes. While the "thinking score" is a good first step, providing concrete examples of successful and unsuccessful reasoning chains would offer deeper insights into the models' strengths and weaknesses.

**Questions:**

1. Could you elaborate on the novelty of WhereBench in the context of recent works like GeoChain and GeoReasoner? A direct comparison would help to clarify your paper's unique contributions.

2. Given the limited size of the dataset, have you considered any statistical methods to validate the significance of your findings?

3. Could you clarify the source of the images used in the WhereCountry dataset?

4. Could you point to common failure modes of some models?

---

> ### Author Response · Authors · 2025-11-27
>
> ### W1. Q1.
> >Marginal Novelty and Missing Citations: The paper's novelty is somewhat diminished by its close resemblance to recent works like GeoChain and GeoReasoner. These papers explore similar themes of geolocation reasoning with VLMs, yet they are not cited in the related works section. Acknowledging and differentiating this work from these existing papers is crucial for establishing its unique contribution.
>
> >Could you elaborate on the novelty of WhereBench in the context of recent works like GeoChain and GeoReasoner? A direct comparison would help to clarify your paper's unique contributions.
>
> We thank the reviewer for identifying these references. We have added GeoChain and GeoReasoner in the revised manuscript Section 2 and included a full comparison table.
> However, we respectfully point out that WhereBench is fundamentally distinct in design, difficulty, and granularity compared to these works. While they share the "geolocation" theme, they serve different roles. We invite the reviewer to examine General Response section 3, where we detail these differences. We believe these distinct contributions—expert verification, street-level granularity, and process analysis—establish WhereBench as a necessary complement to previous benchmarks.
>
> ### W2. Q2.
> >Limited Scale of the Dataset: While well-designed, the WhereBench dataset is relatively small, with only 500 images for WhereCountry and 310 for WhereStreet. This limited scale raises concerns about the statistical significance and generalizability of the findings. For a benchmark paper, a larger and more diverse dataset would be necessary to provide a more robust and reliable evaluation of model performance.
>
> >Given the limited size of the dataset, have you considered any statistical methods to validate the significance of your findings?
>
> We appreciate the reviewer's focus on statistical concern. We have applied 95% Wilson Confidence Intervals to the overall accuracy performance in the revised manuscript Section 4. Our analysis shows that the average 95% confidence interval half-width across the benchmark is $\pm 3.29\%$.
> We invite the reviewer to see the General Response section 1 for the full breakdown of these statistics. We believe these rigorous error bars confirm that WhereBench is a reliable diagnostic tool for the VLM performance.
>
> ### W3.
> >Flawed Assumption about Web Search: The finding that web search does not consistently improve accuracy is presented as a surprising result. However, this is expected given that current web search APIs used by VLMs do not support visual search (i.e., searching with the image itself). The models are limited to searching with textual keywords derived from the image. If these keywords are incorrect or insufficient, the web search will only add noise to the context, potentially degrading performance. This limitation of the underlying tool, rather than a deficiency in the models' reasoning, is a more likely explanation for the observed results.
>
> We appreciate the reviewer’s technical insight regarding the mechanism of current web-search APIs. We agree that the "text-only bottleneck" (converting pixels to text queries) is a primary driver of these failures, and we further note that even image-search pipelines, if not carefully configured, can introduce additional noise to the context. Nevertheless, we argue that this limitation reinforces, rather than diminishes, the value of our findings.
> 1. Query Formulation as part of reasoning: Rather than being “merely a tool limitation,” the ability to transform visual evidence into effective search queries is a core competency of an agentic VLM. The model must analyze a visual input (e.g., a specific tree type), recognize it, and convert it into a searchable textual descriptor. If a model fails to generate accurate keywords and instead searches for generic terms (adding noise), this represents a breakdown in vision-language translation—a reasoning failure that WhereBench explicitly aims to measure.
> 2. Evaluating Systems "As-Is": Our benchmark evaluates SOTA models as end-to-end systems deployed to users. Whether the failure stems from the search API or the model's query generation, the empirical reality is that off-the-shelf agentic VLMs currently fail to leverage the web for recognition-centric tasks. This is a critical finding for the community, highlighting that simply "plugging in the internet" is not a magic bullet for geolocation without better visual-search integration.
>
> We have revised Section 4.1 to explicitly discuss this bottleneck. We now separate the analysis of pure reasoning vs. web-enabled settings to clarify that the performance degradation in WhereCountry is likely due to the model's inability to translate sparse visual clues into sufficiently specific textual queries.

---

> > ### Author Response · Authors · 2025-11-27
> >
> > ### W4. Q3.
> > >Dataset Bias and Lack of Transparency: The WhereStreet dataset is heavily skewed towards Asia, which could introduce regional biases into the evaluation. Additionally, the paper is unclear about the source of the images used in the WhereCountry dataset. Providing more transparency about the data sourcing and composition would improve the benchmark's credibility.
> >
> > >Could you clarify the source of the images used in the WhereCountry dataset?
> >
> > We appreciate the reviewer’s attention to data composition. As detailed in Section 3.1 and Appendix C.1, the WhereCountry images are sourced from the GeoComp dataset[1]. GeoComp consists of worldwide Google Street View images with real human gameplay data. Constructing our benchmark from this source guarantees the locatability of each image. Regarding the dataset bias, we take the concern seriously. To further balance the distribution, we are expanding our benchmark with additional data as detailed in the General Response section 1. We will update the statistics once it is finished.
> >
> > [1]Song, Zirui, et al. "Geolocation with real human gameplay data: A large-scale dataset and human-like reasoning framework." arXiv preprint arXiv:2502.13759 (2025).
> >
> > ### W5. Q4.
> > >Lack of Qualitative Analysis: The paper would be significantly strengthened by a more in-depth qualitative analysis of the models' reasoning processes. While the "thinking score" is a good first step, providing concrete examples of successful and unsuccessful reasoning chains would offer deeper insights into the models' strengths and weaknesses.
> >
> > >Could you point to common failure modes of some models?
> >
> > We appreciate the suggestion to deepen the qualitative analysis. As detailed in Section 4.4 and Appendix E, we have identified three primary failure modes in current VLMs: **Overthinking**: Models correctly identify the location initially but then hallucinate contradictory reasons to reject their own correct hypothesis. **Search Inefficiency**: Models fail to translate visual features into effective search queries. **Underuse Visual Clues**: With web search enabled, models tend to prioritize retrieved text over pixel-level evidence, causing them to overlook visual features.
> > To provide the balanced view requested by the reviewer, we have expanded Appendix E in the revision to include two new successful examples Table 14 and 15, upon the existed failure cases in Table 16 and 17.

---

> ### Comment · Reviewer_A8fX · 2025-11-27
>
> I thank the authors for responding to my questions.
>
> 1. Re related works (GeoChain and GeoReasoner): I agree that these related works rely on discriminatory powers of a few models to determine the difficulty. In theory, replying on human raters would be more representative of difficulty. Do you have any quantitative/qualitative comparisons to show this difference?
>
> 2. Re scale of dataset: Your claims of ~3pp Wilson CI is based on 810 images which is the combined size of both benchmarks. But the individual benchmarks in the datasets (which do not seem to be used in combination in the paper for reporting results) are actually much smaller in size. I expect the CI windows to be much broader for them. And this still remains a point of concern.
>
> 3. Re Web Search: While I agree with the authors that we should be testing the VLMs out of the box as is, it is still not surprising (or a new unintuitive finding) that the inability of recognize a place inhibits the VLMs from using these tools effectively. Infact, the additional noise from the search outputs are more likely to hurt the model's performance - which is in-line with the author's findings.
>
> 4. Re dataset expansion: I appreciate the authors efforts to expand the dataset size and variety. Since this efforts seems to be ongoing and not presented yet, I will reserve my review till this is done.
>
> 5. Re Qualitative Analysis: I appreciate the qualitative analysis provided/added by the authors. It definitely adds insight into to address my comments.

---

> > ### Author Response · Authors · 2025-12-04
> >
> > We thank the reviewer for acknowledging our response to the previous questions. We address the follow-up questions below.
> >
> > ### 1.
> > >Re related works (GeoChain and GeoReasoner): I agree that these related works rely on discriminatory powers of a few models to determine the difficulty. In theory, replying on human raters would be more representative of difficulty. Do you have any quantitative/qualitative comparisons to show this difference?
> >
> > As human annotators do not produce the same type of probability scores as models, we instead qualitatively show three examples in the Appendix for GeoChain and WhereStrees from Tables 18-23. Regarding GeoReasoner, a comparison was not possible as their test set has not yet been released (https://github.com/lingli1996/GeoReasoner/issues/3).
> >
> > ### 2.
> > >Re scale of dataset: Your claims of ~3pp Wilson CI is based on 810 images which is the combined size of both benchmarks. But the individual benchmarks in the datasets (which do not seem to be used in combination in the paper for reporting results) are actually much smaller in size. I expect the CI windows to be much broader for them. And this still remains a point of concern.
> >
> > Thanks for your question. We calculate each model's 95\% confidence interval on WhereStreet and WhereCountry in separate and the results are shown in the table below.
> >
> > | Model                     | Acc (WhereCountry) | CI HW (WhereCountry) | Acc (WhereStreet) | CI HW (WhereStreet) | N (WhereStreet) |
> > |---------------------------|--------------------|----------------------|-------------------|---------------------|-----------------|
> > | Gemini-2.5-pro            | 0.6840             | 0.040620             | 0.3664            | 0.053541            | 307             |
> > | o3 (high)                 | 0.6460             | 0.041772             | 0.3477            | 0.052757            | 309             |
> > | Gemini-2.5-pro (search)   | 0.6446             | 0.041824             | 0.4059            | 0.054354            | 310             |
> > | o3 (high, search)         | 0.6421             | 0.041876             | 0.3498            | 0.052747            | 310             |
> > | GPT4o                     | 0.6160             | 0.042478             | 0.3105            | 0.051702            | 305             |
> > | GPT5 (high, search)       | 0.6097             | 0.042598             | 0.3925            | 0.054105            | 309             |
> > | GPT5 (high)               | 0.6082             | 0.042637             | 0.3344            | 0.052404            | 308             |
> > | o4-mini (high, search)    | 0.5091             | 0.043651             | 0.3219            | 0.052901            | 297             |
> > | GPT4o (search)            | 0.4845             | 0.043637             | 0.2930            | 0.050574            | 309             |
> > | o4-mini (high)            | 0.4645             | 0.043547             | 0.2135            | 0.045427            | 310             |
> > | GLM-4.5V                  | 0.4380             | 0.043325             | 0.1999            | 0.044531            | 309             |
> > | Gemini-2.5-flash (search) | 0.4176             | 0.043073             | 0.3328            | 0.052153            | 310             |
> > | Gemini-2.5-flash          | 0.4000             | 0.042784             | 0.2664            | 0.049071            | 310             |
> > | Claude 4 Opus             | 0.2920             | 0.039734             | 0.2124            | 0.045944            | 302             |
> > | Claude 4 Sonnet           | 0.2760             | 0.039070             | 0.1838            | 0.043484            | 304             |
> > | Skywork-R1V3              | 0.2305             | 0.036804             | 0.1103            | 0.035540            | 304             |
> >
> > The average CI on WhereCountry is 4.18% on 500 samples and the
> > average CI on WhereStreet is 4.95% on different size of samples reported in the table column N. We believe such CI is small enough and support our conclusion about relative model performance and the impact of web search and reasoning effort.

---

> ### Author Response · Authors · 2025-12-04
>
> ### 3.
>
> >Re Web Search: While I agree with the authors that we should be testing the VLMs out of the box as is, it is still not surprising (or a new unintuitive finding) that the inability of recognize a place inhibits the VLMs from using these tools effectively. Infact, the additional noise from the search outputs are more likely to hurt the model's performance - which is in-line with the author's findings.
>
>
> Several studies[1,2,3] suggests that current VLMs surpass average human performance, creating the impression that VLMs have largely solved this task. In contrast, our work demonstrates that even web-tool enhanced VLMs still fail to outperform human experts on our Wherebench.
> The conclusion contradictory to previous study is very surprising, and we believe it is critical to faithfully reflect SOTA VLMs' performance on such task.
>
>
> [1] Huang, Jingyuan, et al. "Vlms as geoguessr masters: Exceptional performance, hidden biases, and privacy risks." arXiv preprint arXiv:2502.11163 (2025).
>
> [2] Zhang, Gengyuan, et al. "Can vision-language models be a good guesser? exploring vlms for times and location reasoning." Proceedings of the IEEE/CVF Winter Conference on Applications of Computer Vision. 2024.
>
> [3] Liu, Yi, et al. "Image-based geolocation using large vision-language models." arXiv preprint arXiv:2408.09474 (2024).
>
> ### 4.
> >Re dataset expansion: I appreciate the authors efforts to expand the dataset size and variety. Since this efforts seems to be ongoing and not presented yet, I will reserve my review till this is done.
>
> Following the procedure outlined in General Response Section 1, we have continued collecting additional samples from Reddit(https://www.reddit.com/r/geochallenges/). To date, we have screened 355 candidate street view images and curated 34 high-quality samples. The geographic distribution of these new samples is shown below:
> | Continent     | Count | Percentage |
> |---------------|-------|------------|
> | Ocean/Sea     | 12    | 35.29%     |
> | Africa        | 9     | 26.47%     |
> | South America | 6     | 17.65%     |
> | Asia          | 5     | 14.71%     |
> | Oceania       | 2     | 5.88%      |
>
> Incorporating these new samples, the overall data distribution is updated as follows:
> | Continent      | Count | Percentage |
> |----------------|-------|------------|
> | Asia           | 187   | 54.36%     |
> | Europe         | 54    | 15.70%     |
> | Island/Oceania | 64    | 18.60%      |
> | North America  | 17    | 4.94%      |
> | South America  | 10    | 2.91%      |
> | Africa         | 12   | 3.49%      |
> | **Total**      | **344** | **100%** |
>
>
> We then evaluated Gemini-2.5-flash on the Reddit subset and the results are shown below:
>
> | Acc@k | Accuracy (%) |
> |----------|----------|
> | 1 km     | 26.47   |
> | 5 km     | 50.00   |
> | 10 km    | 50.00   |
> | 20 km    | 52.94   |
> | 50 km    | 70.59   |
> | 100 km   | 70.59   |
> | 200 km   | 70.59   |
>
> In comparison to the performance on Bilibili and YouTube, the Reddit subset presents an intermediate level of difficulty for Gemini-2.5-flash. We are currently processing the remaining 660 candidate samples and anticipate including an additional 70-100 high-quality images from underrepresented area such as Africa or America. This will bring the total dataset size to approximately 1,000 samples with a more balanced distribution, at which point we will update the final results table.

---

### Official Review · Reviewer_1bwU · 2025-10-31

**Soundness:** 3
**Presentation:** 3
**Contribution:** 3
**Rating:** 6
**Confidence:** 3

**Summary:**

This paper presents WhereBench, a benchmark to evaluate how well vision–language models can perform visual geolocation and reason about geographic context. The dataset contains 810 images split into two subtasks: WhereCountry (coarse, 500 samples) and WhereStreet (fine, 310 samples). The benchmark measures both localization accuracy and reasoning quality, using human-verified visual clues and a Shapley-based thinking score. Twelve recent models are evaluated, with and without web search. Results show that explicit reasoning and retrieval do not reliably improve localization, and that models exhibit strong regional biases. The work highlights a persistent gap between general reasoning and grounded geographic understanding.

**Strengths:**

Originality: The paper fills a clear gap in multimodal evaluation by targeting geographic reasoning across scales. It complements recent reasoning-focused datasets like GeoChain, which emphasize step-by-step inference. WhereBench instead measures end-to-end performance, providing a complementary perspective.

Quality: The benchmark is carefully designed, with globally balanced sampling, reasoning-clue annotations, and transparent evaluation metrics. The Shapley-weighted thinking score is an interesting way to assess reasoning faithfulness.

Clarity: The paper is well organized, clearly written, and supported by good figures and visual examples. The reasoning and bias analyses are easy to follow.

Significance: The findings reveal key weaknesses in current multimodal systems, especially their bias toward familiar regions and limited grounding in fine-grained cues like vegetation, signage, or lighting. The benchmark will likely serve as a useful testbed for future models aiming for spatial or embodied reasoning.

**Weaknesses:**

The dataset is relatively small (810 samples), which limits the reliability of regional generalization and fine-grained comparisons.

The thinking-score metric relies on LLM judgment, and it is unclear how consistent it is across judge models or prompts. A small human correlation study would strengthen the claim.

The error analysis is mainly qualitative. More quantitative breakdowns (e.g., cue failures, retrieval errors) would improve interpretability.

The discussion of regional bias is insightful but short. It would help to quantify which regions or cultural features cause the largest degradation.

Some of the results are descriptive rather than hypothesis-driven. Clearer framing of expected effects from reasoning or retrieval would make the conclusions stronger.

**Questions:**

1. How reproducible is the Shapley-based thinking score across different LLM judges or seeds?
2. Did you measure agreement between human reasoning annotations and model thinking scores?
3. Could reasoning-supervised datasets like GeoChain help improve model performance on WhereBench?

---

> ### Author Response · Authors · 2025-11-27
>
> ### W1
> >The dataset is relatively small (810 samples), which limits the reliability of regional generalization and fine-grained comparisons.
>
> Thank you for your comments. We kindly refer you to the General Response section 1, where we detail the statistical significance of our dataset (average confidence interval $\pm 3.29\%$) and justify our design choice to prioritise high-fidelity, expert-verified reasoning over noisy large-scale data.
> ### W2. Q1. Q2.
> >The thinking-score metric relies on LLM judgment, and it is unclear how consistent it is across judge models or prompts. A small human correlation study would strengthen the claim.
> > How reproducible is the Shapley-based thinking score across different LLM judges or seeds?
> > Did you measure agreement between human reasoning annotations and model thinking scores?
>
> We appreciate your concern regarding the metric's reliability. We kindly refer you to the General Response section 2, where we clarify that Shapley weights are pre-computed and determined, ensuring consistent evaluation, and we include new experiments showing human-model and model-model agreement on Shapley weighted thinking score.
>
> ### W3.
> >The error analysis is mainly qualitative. More quantitative breakdowns (e.g., cue failures, retrieval errors) would improve interpretability.
>
> Thanks for your suggestion. We have updated the manuscript (Section 4.4, Appendix D, and Table 12) to include a systematic quantitative error analysis. We follow the three distinct failure modes and apply LLM-as-a-judge on five leading closed and open-weight models.
>
> | Model                   | Missed Visual Clues (%) | Overthinking (%) | Incomplete Search Error (%) |
> |-------------------------|------------------------:|-----------------:|---------------------------:|
> | o4-mini (search,high)     | 52.19                   | 1.35             | 49.49                      |
> | o3 (high)                 | 43.04                   | 2.91             | 38.51                      |
> | Gemini-2.5-pro          | 37.10                   | 0.97             | 54.84                      |
> | Gemini-2.5-pro (search) | 35.16                   | 2.90             | 54.52                      |
> | GLM-4.5V                | 63.43                   | 34.95            | 7.44                       |
> | Skywork-R1V3            | 76.97                   | 35.86            | 48.68                      |
>
> From the results, though current closed SOTA models perform better at visual grounding and reasoning stability than open-weighted models, they struggle to search queries correctly. Furthermore, open-weight models like Skywork-r1v3 and GLM-4.5-V suffer from a specific "Overthinking" error (~35%), often hallucinating complex rationales that drift away from the visual ground truth.
>
> ### W4.
> >The discussion of regional bias is insightful but short. It would help to quantify which regions or cultural features cause the largest degradation.
>
> We thank the reviewer for encouraging a deeper analysis of regional effects. However, on further reflection, we realized that this framing risks overstating what our current data can support: disentangling the influence of training data, model origin, and benchmark composition would require much more controlled evidence than we have. In the interest of precision and avoiding speculative causal claims, we have removed this specific interpretation from the revision.
> ### W5.
> >Some of the results are descriptive rather than hypothesis-driven. Clearer framing of expected effects from reasoning or retrieval would make the conclusions stronger.
>
> We appreciate the reviewer’s suggestion and have clarified our framing accordingly. Our experiments were guided by the natural hypothesis that stronger reasoning effort and access to web search should consistently improve geolocation accuracy. However, our results do not support this expectation: in several settings, additional reasoning or retrieval yields limited gains, and web search can even harm performance. This suggests that current models may also be bottlenecked by visual perception and clue extraction, besides a lack of reasoning steps or external information. We have revised Section 4 to make this hypothesis and its unexpected empirical outcomes more explicit.

---

> > ### Author Response · Authors · 2025-11-27
> >
> > ### Q3.
> > >Could reasoning-supervised datasets like GeoChain help improve model performance on WhereBench?
> >
> > Thanks for your question. We believe extra finetuning on the geolocation task will improve model performance on our benchmark. As GeoChain is a dataset without any finetuned model, we evaluated GeoReasoner[1], which uses pretrained Qwen-VL and finetuned on the proposed GeoReasoner dataset, and the results are shown below:
> >
> > | Metric | GeoReasoner | Qwen-VL |
> > | :--- | :--- | :--- |
> > | **WhereCountry** | 25.60% | 22.80% |
> > | **WhereStreet-Answer** | 7.66% | 7.58% |
> > | **WhereStreet-Thinking** | 0.133 | 0.164 |
> > | **WhereStreet-Acc@1km** | 3.84% | N/A |
> > | **WhereStreet-Acc@20km** | 21.53% | N/A |
> >
> > Note, Qwen-VL is only able to respond to 7.8% of all WhereStreet coordinate samples. Thus, we report it as 'N/A' in the Acc@k case. From the results, additional training could bring some benefits on WhereCountry, but demonstrate the same-level performance on the harder WhereStreet subset. This also reveals the complex ability requirements on WhereStreet, not a mere dataset distribution difference.
> >
> > [1]Li, Ling, et al. "Georeasoner: Geo-localization with reasoning in street views using a large vision-language model." Forty-first International Conference on Machine Learning. 2024.]

---

### Official Review · Reviewer_YGZR · 2025-11-01

**Soundness:** 3
**Presentation:** 3
**Contribution:** 2
**Rating:** 4
**Confidence:** 4

**Summary:**

WhereBench is a benchmark with 810 images designed to evaluate Vision Language Models (VLMs) on image-based geolocation capabilities. It consists of two tasks - WhereCountry and WhereStreet. WhereCountry consists of 500 country-level identification MCQ problems, and WhereStreet comprises 310 street-level identification questions that are evaluated on final answer and a multi-step reasoning process. The WhereCountry samples were filtered from an initial pool of 8,041 images from GeoComp dataset using an open-source model to remove simple case and low-information images. The WhereStreet samples were extracted frames from 503 public geolocation videos, transcribed by Gemini-2.5-Pro to identify 861 visual clues, and manually verified by 7 PhD volunteers. The authors use a novel Shapley-reweighted thinking score that evaluates the model's intermediate reasoning process to check if it uses human provided visual clues. WhereBench was evaluated on 12 VLMs with Gemini-2.5-Pro achieving the best performance. Results also show that web search and deeper reasoning do not consistently improve performance on WhereCountry task but provide a small gain on WhereStreet task.

**Strengths:**

1. The evaluation method proposed by authors is an interesting way of quantifying how a model uses visual evidence.
2. The selection of MCQ options using UN geoscheme and UN regional groups is clever for making the choices harded for WhereCountry.
3. The benchmark design allows us to compare a variety of models including tool calls and web search models.

**Weaknesses:**

1. The evaluation pipeline relies heavily on Gemini-2.5-Pro for estimating the Shapely values for all clue subsets. This makes the evaluation expensive and limits its reproducibility.
2. While the related works section talks briefly about geolocation benchmarks, the section feels incomplete and does not position WhereBench against other works. The authors should use this section to motivate the problem better.
3. Most models are able to achieve very high performance on this task which makes me question about the need of this benchmark.

**Questions:**

1. What was the annotation process for the 861 key visual clues?
2. The filtering process tries to remove simple examples, could this have introduced a bias?

---

> ### Author Response · Authors · 2025-11-27
>
> ### W1
> > The evaluation pipeline relies heavily on Gemini-2.5-Pro for estimating the Shapely values for all clue subsets. This makes the evaluation expensive and limits its reproducibility.
>
> Thank you for your comments. We kindly refer you to the General Response section 2, where we address the Shapley evaluation concerns as each clue combination is precalculated by prompting Gemini-2.5-pro. Thus, a standard Shapley clue value is used during evaluation and no recalculation is needed.
>
> ### W2
> > While the related works section talks briefly about geolocation benchmarks, the section feels incomplete and does not position WhereBench against other works. The authors should use this section to motivate the problem better.
>
> Thank you for your thoughtful comments. We kindly refer you to the edited manuscript and General Response section 3. We include additional related works and a comparison table to better demonstrate the novelty of our WhereBench.
>
> ### W3
> > Most models are able to achieve very high performance on this task which makes me question about the need of this benchmark.
>
> We appreciate the reviewer’s observation. However, while models can achieve high performance on specific subsets (e.g., coarse-grained regions or Western-centric data), our benchmark reveals that this capability does not generalize robustly on finer-grained localization tasks. On examples that require extensive reasoning and effective tool use, model performance degrades substantially. For instance, Gemini-2.5-pro achieves only 6.38% Acc@1km on Bilibili subset, and GPT-5 achieves 28.1% on textual answers.
> *Overall SOTA is still low.* We include an overall performance table in our edited manuscript section 4, and the best closed-source model achieves only 56.32%, and open-weighted model achieves only 34.89%.  This leaves a roughly 45% error rate even for SOTA models.
> *Process is over Results.* WhereBench is designed to evaluate **how** models solve tasks, not just if they solve them. Our analysis shows that models often get the right answer for the wrong reason. Understanding these failure modes is critical for future development, regardless of raw accuracy scores.
> ### Q1
> >What was the annotation process for the 861 key visual clues?
>
> Thanks for the question regarding the annotation procedure. We have updated Appendix C.2 to fully detail the annotation protocol. In summary, the 861 key clues were extracted and summarized by an LLM from videos and then rigorously verified by 7 trained annotators against the raw footage. We strictly filtered for observable visual evidence (e.g., road markings, vegetation) and removed any non-visual deductions. Ambiguous cases were decided by a lead geolocation expert to ensure high quality.
>
> ### Q2
> >The filtering process tries to remove simple examples, could this have introduced a bias?
>
> We appreciate the reviewer’s query regarding potential bias. We acknowledge that our filtering process alters the data distribution compared to random sampling; however, this is an *intentional design choice* to ensure the benchmark measures deductive reasoning rather than simple recognition. Raw Google Street View image distributions are frequently dominated by "trivial" samples solvable purely via OCR, such as reading unique merchant names or road signs, which allow models to bypass complex visual analysis entirely. By explicitly filtering these out, as a standard procedure in many geolocation benchmarks [1,2,3] to avoid dataset saturation, we prevent "ceiling effects" where models appear competent due to shortcuts. This ensures that WhereBench retains high discriminative power, reflecting genuine robust geolocation capabilities.
>
> [1] Li, Ling, et al. "Georeasoner: Geo-localization with reasoning in street views using a large vision-language model." Forty-first International Conference on Machine Learning. 2024.
>
> [2] Campos, Ron, et al. "Gaea: A geolocation aware conversational model." arXiv e-prints (2025): arXiv-2503.
>
> [3] Li, Lingyao, et al. "From pixels to places: A systematic benchmark for evaluating image geolocalization ability in large language models." arXiv preprint arXiv:2508.01608 (2025).

---

### Author Response · Authors · 2025-11-27

## General Response
We are grateful for the reviewers’ thoughtful and constructive feedback on our submission. We are encouraged by the recognition of the importance of geolocation-focused multimodal reasoning, the careful design of our benchmark and evaluation protocol, and the novelty of our Shapley-reweighted thinking score for assessing reasoning faithfulness. We now address the three common concerns in the sections below.
### Q1. Dataset Size

> Reviewer 1bwU: The dataset is relatively small (810 samples), which limits the reliability of regional generalization and fine-grained comparisons.

> Reviewer A8fX: While well-designed, the WhereBench dataset is relatively small, with only 500 images for WhereCountry and 310 for WhereStreet. This limited scale raises concerns about the statistical significance and generalizability of the findings. For a benchmark paper, a larger and more diverse dataset would be necessary to provide a more robust and reliable evaluation of model performance.

> Reviewer iQxJ: The benchmark dataset is extremely small, with only 810 total examples (500 for WhereCountry, 310 for WhereStreet), making it difficult to draw statistically significant or generalizable conclusions.

We appreciate the reviewers’ concerns regarding dataset size and agree that larger and more diverse benchmarks are beneficial in general. At the same time, we would like to clarify that WhereBench is intentionally designed as a **high-quality hard geolocation benchmark with global coverage**, and that its current scale already supports statistically meaningful comparisons across models.

**First, our 810 images are the result of an aggressive curation pipeline designed to produce challenging, representative geolocation tasks while maintaining worldwide coverage.** For WhereCountry, we start from 8,041 GeoComp images and filter out “easy” cases (e.g., clear national flags, distinctive text that can be trivially solved by OCR) using open-weight VLMs and human validation, then retain the top 500 images based on human gameplay scores to ensure difficulty. For WhereStreet, we collect 503 full geolocation videos, then manually verify each candidate sample and annotate only those where the location can be deduced from visual evidence, resulting in 310 hard, step-by-step reasoning instances with human-verified key clues.

Such careful selection is crucial for evaluating vision-grounded reasoning and tool use, especially when web search is available. A much larger but less curated test set (e.g., with trivially solvable or visually uninformative images) would risk overestimating model performance.

**Second, with 810 images, it could reach statistical significance.** To directly address concerns about statistical reliability, we have added an overall performance table with 95% Wilson confidence intervals. We include this revised section in our edited manuscript in Sec4. The average 95% CI half-width is ± 3.29%. The performance gaps we report are magnitudes larger than this margin of error. Thus, our sample size is sufficient to support robust conclusions about relative model performance and the impact of web search and reasoning effort.

**Third, WhereBench’s test size is consistent with standard evaluations in this specific domain.** For example, IM2GPS[1], LLMGeo[2], and Fairlocator[3] have 237, 1000, and 1200 test set images, respectively. Our benchmark’s 810 carefully curated and globally distributed images are thus within the common range for test benchmarks in image geolocation, while additionally providing human-verified reasoning annotations and process-level metrics, which are absent in larger, coordinate-only datasets.

We agree that extending WhereBench further is a valuable direction, and we are **working to expand the dataset** by curating additional WhereStreet samples on publicly shared GeoGuessr games on Reddit. For each sample, we extract the corresponding Google Street View panorama and coordinates. Annotators then read the players' solution and distill it into a concise set of key, image-grounded visual clues that explain how the final location is determined. We will update the statistics once the annotation is finished and evaluated on the leading models. However, we maintain that the current version-comprising 810 hard, globally covered, and expert-verified instances-already yields tight confidence intervals and supports statistically robust conclusions regarding VLM geolocation, reasoning efficacy, and tool utilization.

[1] Hays, James, and Alexei A. Efros. "Im2gps: estimating geographic information from a single image." 2008 ieee conference on computer vision and pattern recognition. IEEE, 2008

[2] Wang, Zhiqiang, et al. "Llmgeo: Benchmarking large language models on image geolocation in-the-wild." arXiv preprint arXiv:2405.20363 (2024)

[3] Huang, Jingyuan, et al. "Vlms as geoguessr masters: Exceptional performance, hidden biases, and privacy risks." arXiv preprint arXiv:2502.11163 (2025).

---

> ### Author Response · Authors · 2025-11-27
>
> ### Q2. Shapley-weighted Thinking Score
> >Reviewer YGZR: The evaluation pipeline relies heavily on Gemini-2.5-Pro for estimating the Shapely values for all clue subsets. This makes the evaluation expensive and limits its reproducibility.
>
> >Reviewer 1bwU: How reproducible is the Shapley-based thinking score across different LLM judges or seeds?
> >Reviewer 1bwU: Did you measure agreement between human reasoning annotations and model thinking scores?
>
> >Reviewer iQxJ: The methodology for the novel Shapley-reweighted score is underspecified. The paper states is estimated by "prompting the judge (Gemini-2.5-Pro) for all subsets" but never states the average number of clues , which could make this computationally infeasible. The value function  ("achievable answer quality") is also highly subjective and not clearly defined.
> >Reviewer iQxJ: The paper validates its answer-scoring judge but provides no such validation for its clue-in-reasoning judge. It is unclear how the judge reliably distinguishes a clue that was truly "used" from one that was merely "mentioned."
> >Reviewer iQxJ: How exactly is the "achievable answer quality" score for  defined and prompted to the judge?
>
> We address reviewer questions regarding the consistency, reproducibility, and human alignment of our Shapley-reweighted thinking score.
>
> **Efficiency** We'd like to clarify that the Shapley-weighted scores are computed once per sample and stored as static weights for the benchmark. That is, for each image and its associated clue set, we estimate the Shapley value of each clue (under a fixed judge model and fixed prompts) and store these weights. The thinking score for any model is then obtained by combining these fixed Shapley weights with the model’s revealed clues, without re-running the Shapley procedure. The total cost for obtaining the Shapley-weight is less than $25 with the Gemini-2.5-pro API. We have made this explicit in the revised manuscript to avoid the impression that Shapley values are recomputed per model.
>
> **Reproducibility & Consistency** To measure the intrinsic stability of the Shapley calculation itself, we conducted a variance analysis on 50 randomly selected samples. We re-computed the full Shapley vector 10 times per sample with different seeds on two judges (Gemini-2.5-pro and GPT-5-mini). The resulting Shapley scores showed low variability across runs, with an average standard deviation of 0.068 for Gemini-2.5-pro and 0.061 for GPT-5-mini. Cohen's $\kappa$ agreement between the two models on the induced ordinal rankings of clue subsets is 0.601. This confirms that the value function provides stable ordinal rankings of clue subsets across different models and seeds.
>
> **Value Function** The value function used for Shapley is instantiated via a rigorous LLM-as-a-judge protocol, specified in the prompt in Table 8. For each subset of clues $S$, the judge first determines an anchor score corresponding to how well an ideal solver could answer the task if restricted to only $S$ clues, using the same scoring scale as for hierarchical text answer evaluation. Then, we allow the judge to perturb the final score around the anchor to reflect extra or missing information in one clue combination. In this way, the value function is an ordinal quality score defined consistently for all subsets, and it is applied uniformly across all samples.
>
> **Human-Model Agreement on Thinking Score** To validate the faithfulness of the reasoning, annotators performed a binary verification on a random 50 samples to determine whether each clue was genuinely included in the model's response. This would create vectors similar to $s_i$ defined in Equation (5) in section 3.3. We calculate the Cohen's $\kappa$ of 0.754, confirming the reliability of applying LLM-as-a-judge on deciding clue attribution.
> To validate Shapley weights, we conducted a human-model agreement study on 50 randomly selected subsets. Human experts ranked the marginal contribution of clues, and we compared this score to the Shapley-derived importance. We achieved a Cohen's $\kappa$ of 0.695, demonstrating strong alignment between the human and model's agreement on clue importance.

---

> > ### Author Response · Authors · 2025-11-27
> >
> > ### Q3. Related Work and Novelty
> > >Reviewer YGZR: While the related works section talks briefly about geolocation benchmarks, the section feels incomplete and does not position WhereBench against other works. The authors should use this section to motivate the problem better.
> >
> > >Reviewer A8fX: The paper's novelty is somewhat diminished by its close resemblance to recent works like GeoChain and GeoReasoner. These papers explore similar themes of geolocation reasoning with VLMs, yet they are not cited in the related works section. Acknowledging and differentiating this work from these existing papers is crucial for establishing its unique contribution.
> >
> > >Reviewer iQxJ: The central claim that "there remains a lack of a fair and comprehensive benchmark that evaluates... faithfulness" is incorrect. Several such benchmarks have been proposed at a much larger scale, including GeoChain (Yerramilli et al., 2025) and Gaea (Campos et al., 2025), which the paper fails to cite or compare against.
> >
> > We thank the reviewers for identifying these critical missing references. We have expanded Section 2 in the revised manuscript and have included a feature comparison table to directly compare WhereBench with previous works. That said, we believe WhereBench still offers meaningful contributions beyond these existing efforts. In particular, we distinguish WhereBench along three key dimensions:
> >
> > 1. **Hard street-level reasoning with human verification.**
> > GeoChain and GeoReasoner rely on an automated locatability score (derived from GeoCLIP [1]) to filter images. As a result, the validity of these benchmarks is closely tied to the performance of GeoCLIP itself: when GeoCLIP fails to detect a subtle clue, an image may be miscategorized or unintentionally filtered out.
> > WhereBench takes a different approach. Rather than defining difficulty based on the limitations of current discriminative models, our “hard” samples are drawn from videos in which human experts successfully determine the location using complex, multi-step reasoning—such as combining sun-position cues with region-specific vegetation patterns—that GeoCLIP often overlooks. This allows our benchmark to more faithfully capture the upper bound of expert-level reasoning.
> >
> > 2. **Semi-automatic vs fully automatic pipeline on reasoning evaluation.**
> > Prior works largely derive supervision automatically: GeoChain’s 21 fixed questions and answers are generated from semantic segmentation labels and city-level metadata, and GAEA-Bench’s geographically grounded QA pairs are built from OpenStreetMap attributes within 1km of the coordinates. In both cases, many “clues” may not correspond to what is actually visible in a single image. For example, in GAEA-Bench, one prompt is "Is there an Italian restaurant near the location in this image? If so, can you name it?" but the input image is a bus station without any clue about that Italian restaurant.
> > In contrast, WhereStreet starts from human gameplay on videos and then manually extracts and verifies key clues so that each annotated clue is (i) actually used by humans to solve the location and (ii) visually deducible from the final frame. This ensures that our "Thinking Score" measures actual vision-language alignment and reasoning faithfulness, providing a level of reliability that automated pipelines cannot achieve.
> >
> > 3. **Finer granularity and explicit evaluation of agentic tool use.**
> > GeoReasoner mainly reports country- and city-level performance and uses its curated dataset to train a specific LVLM, rather than serving as a model-agnostic benchmark. GAEA-Bench targets conversational QA about locations and nearby PointsOfInterest derived from OpenStreetMaps metadata, not necessarily pinpointing a precise street from image evidence.
> > WhereBench targets the finest granularity: precise street-level localization and exact coordinates. Such a goal requires significantly more fine-grained analysis of visual evidence than identifying a city or region. Moreover, WhereBench explicitly evaluates agentic behavior: we compare models both with and without web search, directly probing how they apply external tools to connect visual evidence to precise street coordinates, a dimension largely absent in benchmarks focused on static or coarse-grained classification. For example, Figure 1 illustrates how models are expected to iteratively use web search to refine their hypotheses from a coarse region down to an exact street.
> >
> > In summary, WhereBench is a high-precision diagnostic benchmark built from human-verified samples and clues, designed to accurately evaluate VLM geo-guessing performance across complex visual perception, reasoning, and tool-use capabilities, without the noise typically introduced by automated data generation.
> >
> > [1] Vivanco Cepeda, V., Nayak, G. K., & Shah, M. (2023). Geoclip: Clip-inspired alignment between locations and images for effective worldwide geo-localization. Advances in Neural Information Processing Systems, 36, 8690-8701.

---

### Meta-Review · Area_Chair_3bSr · 2025-12-24

**Summary:**

All reviewers acknowledged the novelty of the proposed "thinking score" but assigned scores below the acceptance threshold due to concerns about scale and novelty. Reviewers 1bwU, A8fX, and iQxJ argued that the dataset size (810 samples) is insufficient for statistical significance. Reviewers A8fX, iQxJ, and YGZR questioned the paper's contribution, noting a lack of comparisons with existing large-scale benchmarks like GeoChain. Additionally, Reviewers YGZR and iQxJ raised concerns regarding the reproducibility and cost of the evaluation pipeline. The authors' rebuttal did not sufficiently resolve these core issues.

**Reviewer Concerns:**

The authors responded by adding missing baselines, conducting additional error analyses, and providing confidence intervals. However, these revisions appeared insufficient to address the core concerns. Reviewers remained unconvinced regarding the limited dataset size, noting that the error margins for sub-tasks were still too broad and the ongoing dataset expansion could not be evaluated.

**Reviewer Scores:**

The reviewers did not increase their scores following the rebuttal. Reviewer iQxJ explicitly decided to maintain the original score, and Reviewer A8fX indicated that while the proposed dataset expansion is promising, the review must be based on the current state of the submission, which remains limited.

---

### Decision · Program_Chairs · 2026-01-26

Reject